＃ nature communications

# Capturing dynamical correlations using implicit neural representations

Sathya R. Chitturi ®[1,2] ✉, Zhurun Ji ®[3,4] ✉, Alexander N. Petsch ®[1,5,6] ✉, Cheng Peng[5], Zhantao Chen[5], Rajan Plumley[1,5,7], Mike Dunne[1], Sougata Mardanya ®[8], Sugata Chowdhury[8], Hongwei Chen ®[9], Arun Bansil ®[9], Adrian Feiguin ®[9], Alexander I. Kolesnikov ®[10], Dharmalingam Prabhakaran ®[11], Stephen M. Hayden ®[6], Daniel Ratner[1], Chunjing Jia ®[1,5,12], Youssef Nashed[1] & Joshua J. Turner ®[1,5] ✉

Understanding the nature and origin of collective excitations in materials is of fundamental importance for unraveling the underlying physics of a many-body system. Excitation spectra are usually obtained by measuring the dynamical structure factor, $S(\mathbf{Q}, \omega)$, using inelastic neutron or x-ray scattering techniques and are analyzed by comparing the experimental results against calculated predictions. We introduce a data-driven analysis tool which leverages 'neural implicit representations' that are specifically tailored for handling spectrographic measurements and are able to efficiently obtain unknown parameters from experimental data via automatic differentiation. In this work, we employ linear spin wave theory simulations to train a machine learning platform, enabling precise exchange parameter extraction from inelastic neutron scattering data on the square-lattice spin-1 antiferromagnet $La_2NiO_4$, showcasing a viable pathway towards automatic refinement of advanced models for ordered magnetic systems.

Quantum matter, as featured by the existence of macroscopic orders from microscopic spin and/or charge arrangements or other phases with spontaneous symmetry breaking, represents an abundant and complex class of materials in condensed matter physics. For example, the magnetic configuration of a material and its dynamics are often driven by competing effects of multiple interactions as well as crystalline symmetries. The collective spin excitations in most magnetic materials, such as spin waves or magnons, act as probes of those interactions. The associated dispersion relations and correlations are key for developing potential applications, which include next-generation spintronics devices, as well as new strategies for carrying, transferring, and storing information[1-3].

A primary aim of the last few decades has been to characterize wide classes of excitations, and this has been facilitated by advances in spectroscopic techniques, such as neutron scattering[4-7]. These techniques use the kinematics of scattered neutrons to obtain dispersion relations, lifetimes, and amplitudes of spin excitations. Neutron scattering studies are, however, challenging due to the paucity of available neutron sources, low neutron flux compared to other sources, and small neutron scattering cross sections. As a result, the question of

[1]SLAC National Accelerator Laboratory, Menlo Park, CA 94025, USA. [2]Department of Materials Science and Engineering, Stanford University, Stanford, CA 94305, USA. [3]Department of Physics and Applied Physics, Stanford University, Stanford, CA 94305, USA. [4]Geballe Laboratory for Advanced Materials, Stanford University, Stanford, CA 94305, USA. [5]Stanford Institute for Materials and Energy Sciences, Stanford University, Stanford, CA 94305, USA. [6]H.H. Wills Physics Laboratory, University of Bristol, Bristol BS8 1TL, UK. [7]Department of Physics, Carnegie Mellon University, Pittsburgh, PA 15213, USA. [8]Department of Physics and Astrophysics, Howard University, Washington, DC, USA. [9]Department of Physics, Northeastern University, Boston, USA. [10]Neutron Scattering Division, Oak Ridge National Laboratory, Oak Ridge, TN 37831, USA. [11]Department of Physics, University of Oxford, Clarendon Laboratory, Oxford OX1 3PU, UK. [12]Department of Physics, University of Florida, Gainesville, FL 32611, USA. ✉e-mail: chitturi@stanford.edu; zhurun@stanford.edu; apetsch@stanford.edu; joshuat@slac.stanford.edu

how the efficiency of neutron experiments could be enhanced is drawing considerable interest in the field[8, 9]. Notably, the interpretation of neutron scattering spectra can be challenging and time-consuming due to the complex nature of the physical processes involved, the diversity of samples, and the limited knowledge often provided by theoretical modeling. It is clear that there is an urgent need for collaboration among experiment, theory, and data science to accelerate the understanding of spin-related properties of materials[10].

As the rates of data collection continue to increase rapidly, especially with the advent of next-generation X-ray free electron laser facilities and the ability to collect hyper-dimensional datasets, it is important to develop techniques for real-time modeling and analysis of experimental spectra. The ability to perform 'on-the-fly' fitting[11] would enable efficient use of expensive beamtime by ascertaining when sufficient data has been collected, as well as by coupling to adaptive sampling methods to gain the most information about parameters of interest with the least number of measurements. Currently, real-time fitting for neutron scattering data can require substantial preparation. For example, direct fitting with the software package SpinW[12] requires the extraction of the eigenmodes of the system and therefore, needs an accurate, and preferably automatic, peak extraction algorithm. When the chosen paths in reciprocal space are numerous or the dispersion relations change significantly along those paths, this can involve significant human guidance and monitoring. In addition, fitting directly with SpinW does not take into consideration the magnon peak intensities or their shapes. Approaches to fit peak intensities and shapes directly, such as Multi- or Tobyfit implemented in HORACE[13], are possible alternatives – however, these fitting procedures still require significant human guidance and are either slow and therefore incompatible with data acquisition rates or else they require an analytical, rapidly calculable spin wave model. Finding such a model is usually only feasible for simpler systems with minimal magnetic frustration or a low number of magnetically distinct sites.

Machine learning methods have recently been utilized in the analysis of x-ray and neutron scattering measurements to improve the accuracy and speed of data interpretation[10,14]. Convolutional neural networks, trained on linear spin wave simulations, have been applied to inelastic neutron scattering measurements to discriminate between two plausible magnetic exchange models[14,15]. As these models are typically trained on simulated profiles, achieving robust prediction generally requires detailed modeling and corrective dataset augmentations of experimental effects accounting for attributes such as background noise, missing data, and matching instrumental profiles[15–17]. Recently, a cycleGAN approach, which makes experimental data look like simulated data has been proposed as a way to improve model robustness[16]. In cases where the desired observables have continuous values, such approaches are often highly sensitive to background noise and other effects[18]. To predict continuous Hamiltonian parameters from static and inelastic neutron scattering data, previous approaches have utilized a combination of an autoencoder neural network, used for data compression, and a generative model, used for forward prediction[14,19–21]. This pipeline has been shown to return excellent results on fully collected data but has not previously been applied to the setting of on-the-fly parameter extraction.

Prior machine learning efforts in the neutron scattering community have relied on traditional image-based data representations. A promising direction in this field can be capitalized on with the introduction of a new paradigm of data modeling based on neural implicit representations[22,23]. Such models are often described as coordinate networks as they take a coordinate as input and typically output a single scalar or a small set of scalars. In computational imaging applications, these networks learn mappings from pixel coordinates $(i, j)$ to an RGB value representing the color of that pixel. The coordinate-based representation encodes the image implicitly through a set of trainable weights and can be used to make predictions at sub-pixel scales. These models have been shown to be able to accurately capture high-frequency features in images and scenes and have been particularly successful at tasks such as 3D-shape representation and reconstruction. Furthermore, gradients and higher-order derivatives of the implicit representation can be readily calculated and used for solving inverse-problems[22,24–26].

In this work, we develop a neural implicit representation for the dynamical structure factor, $S(\mathbf{Q}, \omega)$, as a function of energy transfer ($\hbar\omega$), momentum transfer ($\mathbf{Q}$), and Hamiltonian parameter coordinates. The dynamical structure factor is a general function measured in many inelastic x-ray and neutron experiments and is related to different correlation functions of the probed order, see "Methods" section for further details. To demonstrate the versatility of our method, we report the results using a series of calculations based on mean field theory through the linear spin wave theory (LSWT) framework[27]. We simulate LSWT spectra for a spin-1 square-lattice Heisenberg model Hamiltonian over a large phase space of Hamiltonian parameters and use it to train a neural implicit representation. The model is applied to experimental time-of-flight neutron spectroscopy data[28] taken on the quasi-2D Néel antiferromagnet $La_2NiO_4$, and leverages a GPU-based optimization procedure to return the Hamiltonian parameters that represent the system under study. In particular, the method does not rely on peak fitting algorithms and performs well under low signal-to-noise ratio scenarios. To gain further insight, we use a Monte-Carlo simulation of the experimental data collection process to demonstrate the potential of our approach for continuous in-situ analysis to provide guidance on when an adequate amount of data has been collected to conclude the experiment. Collectively, these findings pave the way for conducting scattering experiments in a streamlined and efficient manner, and open exciting new avenues to swiftly unravel the parameterization of underlying dynamical models.

## Results

### Neural implicit representation modeling

Our machine learning framework is based on the concept of implicit neural representations which are machine learning models that can be used to store images (or hypervolumes) via trainable network parameters. Accordingly, we develop a neural implicit representation for the dynamical structure factor across different model Hamiltonian parameters. Our Hamiltonian, which corresponds to an extended nearest-neighbor Heisenberg model, is given by Eq. (1)[7,29].

$$\mathcal{H} = J \sum_{\langle i,j \rangle} \hat{\mathbf{S}}_i \cdot \hat{\mathbf{S}}_j + J_p \sum_{\langle i,j' \rangle} \hat{\mathbf{S}}_i \cdot \hat{\mathbf{S}}_{j'}, \tag{1}$$

As depicted in Fig. 1a, $J$ and $J_p$ are the first- and second-nearest-neighbor Heisenberg exchange coupling parameters on a square lattice. Thus, for $Q_x$ and $Q_y$, a square-lattice notation is utilized with $\mathbf{a}$ and $\mathbf{b}$ corresponding to the vectors connecting the first nearest neighbors or opposing edges of the square, respectively.

The specific implicit neural representation presented in this work is a Sinusoidal Representation Network (SIREN)[22], which is a fully-connected neural network[30] with sinusoidal activation functions that accepts coordinates as input. Our SIREN model is trained to approximate the scalar function $\log(1 + S(\mathbf{Q}, \omega, J, J_p)) \in \mathbb{R}^1_+$ (a real positive number $\{x \in R | x > 0\}$), which is a logarithmic transformation of the dynamical structure factor evaluated at a specific $\mathbf{Q} \in \mathbb{R}^3$ (3 dimensional, reciprocal lattice vector in reciprocal lattice units (r.l.u.)), $\hbar\omega \in \mathbb{R}^1$ (energy transfer in units of meV) and $J, J_p \in \mathbb{R}^1$ (specific Hamiltonian coupling parameters in units of meV). We use the logarithm to

increase the weighting of weaker features in the data and add one to prevent ill-conditioned behavior around zero.

The functional form of the SIREN neural network, denoted as $\Phi$, involves applying a series of matrix multiplications, vector additions and sinusoidal operations to the coordinate vector $\left[\mathbf{Q}, \omega, J, J_p\right]^{\top} \in \mathbb{R}^6$ (Eq. (2)).

$$
\begin{aligned}
h_0 &= \sin(W_0\left[\mathbf{Q}, \omega, J, J_p\right]^{\top} + b_0) \\
h_i &= \sin(W_i h_{i-1} + b_i) \text{ with } i \in \{1, 2, 3\} \\
\Phi &= (W_4 h_3) + b_4
\end{aligned}
\tag{2}
$$

Here, $b_0 \in \mathbb{R}^6$, $\{b_1, b_2, b_3\} \in \mathbb{R}^{64}, b_4 \in \mathbb{R}^1, W_0 \in \mathbb{R}^{64 \times 6}, \{W_1, W_2, W_3\} \in \mathbb{R}^{64 \times 64}$ and $W_4 \in \mathbb{R}^{1 \times 64}$ are vectors and matrices, respectively, that are learned during the training process to ensure that $\Phi(\mathbf{Q}, \omega, J, J_p)$ mimics $\log(1 + S(\mathbf{Q}, \omega, J, J_p))$ as closely as possible. Graphically, $W_0, W_1, W_2,$ and $W_3$ correspond to the weights between the first four layers of the network which are transformed by applying the sine function in an element-wise manner. $W_4$ represents the weights for the final layer for which only a linear function is applied. This specific neural architecture is also illustrated in Fig. 1c.

We note that although our model is written for three-dimensional $\mathbf{Q}$, the neutron profiles used in the following sections do not include a $Q_z$ component due to limited sample orientations. The model is trained on 1200 LSWT simulations of $S(\mathbf{Q}_{\text{list}}, \omega_{\text{list}})$ over a large set of possible $J$ and $J_p$

values and on two paths in reciprocal space (Fig. 1b). **Q**-path 1 and 2 are denoted as $P \to M \to X \to P \to \Gamma \to X$ and $P1 \to M1 \to X1 \to P1 \to \Gamma1 \to X1$ which correspond to $\mathbf{Q}_{\text{path1}} = \left\{ \left[\frac{3}{4} \frac{1}{4} 0\right], \left[\frac{1}{2} \frac{1}{2} 0\right], \left[\frac{1}{2} 0 0\right], \left[\frac{3}{4} \frac{1}{4} 0\right], [1 0 0], \left[\frac{1}{2} 0 0\right] \right\}$ and $\mathbf{Q}_{\text{path2}} = [-0.07\ 0.03\ 0] + \mathbf{Q}_{\text{path1}}$. Here, $\mathbf{Q}_{\text{list}} \in \mathbb{R}^{N_{\mathbf{Q}}}$ and $\omega_{\text{list}} \in \mathbb{R}^{N_\omega}$ is an overloaded notation which refers to a series of $N_{\mathbf{Q}}$ and $N_\omega$ points in the $(\mathbf{Q}, \omega)$-space, respectively.

Once the differentiable neural implicit model is trained, it is possible to use gradient-based optimization to solve the inverse problem of determining the unknown $J$ and $J_p$ parameters from data. Our objective function for the optimization task measures the Pearson correlation coefficient (r) between $\log(1 + S(\mathbf{Q}, \omega, J, J_p))$ and the machine learning prediction (Equation (3)).

$$
L = 1 - r(\log(1 + S_{\text{measured}}), \Phi(\mathbf{Q}, \omega, J, J_p))
\tag{3}
$$

We use the correlation as the metric because the normalization factors between the experiment and simulation are here unknown. Using the logarithmic transformation is favorable as it enhances the weighting of the coherent excitation at high $\hbar\omega$ and further helps evade contamination due to statistical noise in the elastic and incoherent-inelastic scattering, which arises primarily at low $\hbar\omega$ and that cannot be removed by background subtraction. The normalization scheme is important since we are not aiming to fully describe the spectral weights, which would require the exact handling of all individual neutrons in the full three-

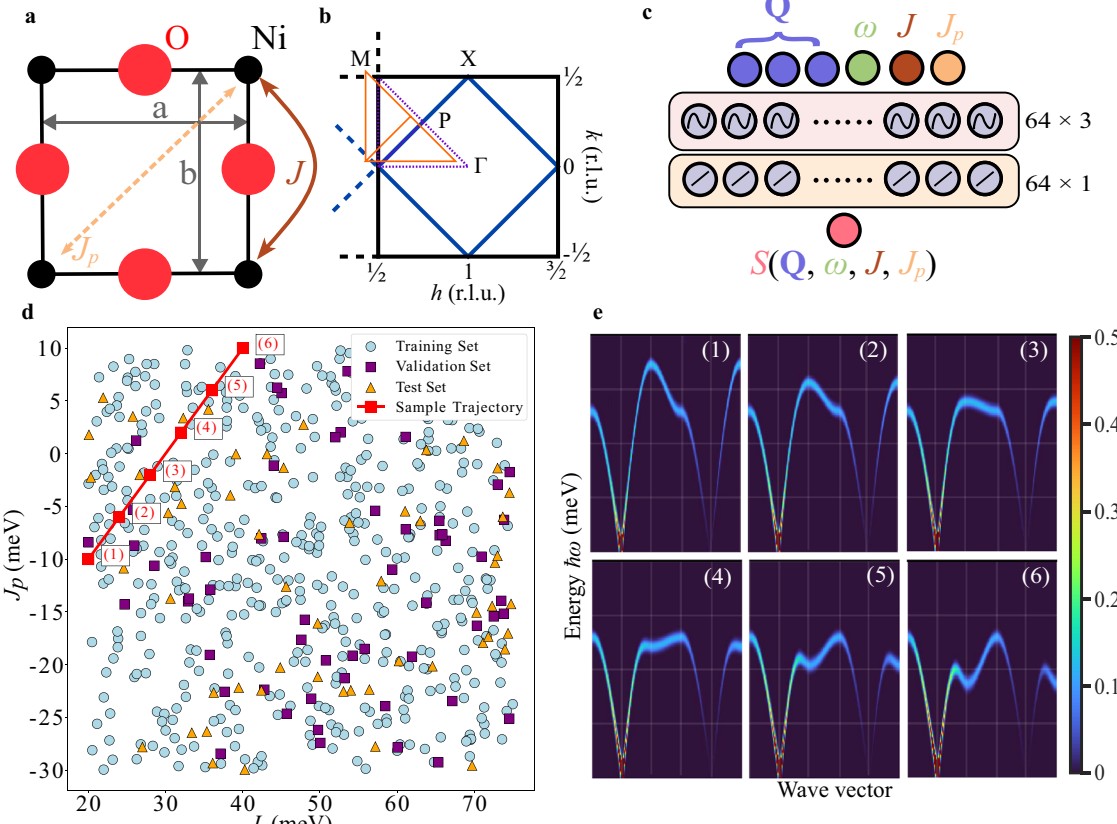

**Fig. 1 | Overview of machine learning pipeline, model Hamiltonian and reciprocal space paths. a** $Ni_4O_4$ square-lattice plaquette in $La_2NiO_4$. $J$ ($J_p$) is the first (second) nearest-neighbor interaction and a and b indicate the square-lattice unit vectors. **b** The Brillouin zone for the spin-1 square-lattice magnetic structure. Selected high-symmetry points are indicated. The two momentum paths are denoted by the purple and orange lines, respectively. **c** Visualization of the SIREN neural network for predicting the scalar dynamical structure factor intensity. All nodes in adjacent layers are connected to each other in a fully-connected architecture. The notation $64 \times 3$ and $64 \times 1$, represent three and one neural network layers with 64 neurons and with sinusoidal and linear activation functions, respectively. Neural network bias vectors are omitted for clarity. **d** Visualization of the distribution of training, test, and validation data in $J$-$J_p$ space. **e** Synthetic $S(\mathbf{Q}, \omega)$ predictions from the SIREN model along the corresponding trajectory shown in **d**. Grid lines correspond to [0, 50, 100, 150, 200] meV and [P, M, X, P, $\Gamma$, X] for the energy and wave vector, respectively.

dimensional **Q**-space, instead of the averaged weight in the reduced two-dimensional **Q**-space. During optimization, any subset of ($\mathbf{Q}_{list}$, $\omega_{list}$) coordinates can be chosen as long as they fall along either of the paths defined in Fig. 1b. Here, we note that from an inference point of view, any momentum or energy coordinates could be chosen, however our training data only includes two reciprocal-space paths. To determine the Hamiltonian parameters, $J$ and $J_p$ are treated as free parameters in the optimization problem. The objective in Eq. (3) is optimized using the Adam optimizer[31], a commonly used gradient-based optimization algorithm that exploits the automatic differentiation capabilities in Tensorflow[32] to calculate $\frac{dL}{dJ}$ and $\frac{dL}{dJ_p}$, see "Methods" section for details.

In our technique, it is not necessary to use all sets of $\mathbf{Q}_{list}$, $\omega_{list}$ along both paths to perform the fitting. Instead, random batches of coordinates ($\mathbf{Q}_{batch}$, $\omega_{batch}$) can be queried at each optimization iteration in order to improve computational efficiency and converge to a better minimum, in a manner similar to the regularization effects of stochastic gradient descent[33]. Pseudo-code for the optimization procedure is provided in Algorithm 1.

**Algorithm 1.** Differentiable Neural Optimization
**while** N < MaxIter **do**
    $\mathbf{Q}_{batch}$, $\omega_{batch}$, $S_{batch}$ ~ [$\mathbf{Q}_{list}$, $\omega_{list}$, $S_{list}$]
    $\log(1 + S_{pred}) = \Phi(\mathbf{Q}_{batch}, \omega_{batch}, J, J_p)$
    $J, J_p \leftarrow$ ADAM(L($S_{batch}$, $S_{pred}$))
**end while**

## Application to La$_2$NiO$_4$

We first characterize the performance of our machine learning framework on simulated SpinW data in order to demonstrate the viability of using a neural implicit representation for the LSWT simulator. Figure 1e demonstrates the ability of our implicit model to generate new predictions for $S(\mathbf{Q}, \omega)$ under Hamiltonian parameter ranges that lie outside the training data. Figure 2 provides a comparison between the LSWT and machine learning simulation with specific values of the input parameters ($J = 45.57$ meV and $J_p = 2.45$ meV). In this example, the machine learning framework was fed ($J, J_p$) directly (instead of obtaining these parameters using gradient descent through the neural representation). The machine learning prediction and the LSWT simulation are seen to be almost indistinguishable. A quantitative analysis of the difference between simulation and prediction is provided in Supplementary Fig. 2.

Although our model can clearly approximate simulated data well, our main motivation, however, is to provide a tool that can reliably extract the spin Hamiltonian parameters of interest from real, experimental data. For this reason, we applied our method to the measured inelastic neutron scattering data (after an automatic background-subtraction) taken from the quasi-2D Néel antiferromagnet La$_2$NiO$_4$. Experimental data prior to background subtraction are shown in Supplementary Fig. 1. Though a full 3D dataset was collected, we chose two paths in **Q**-space to simulate many spectra for a range of $J$ and $J_p$ for the model training prior to any inclusion of real data. After the model was trained on the two simulated paths, we applied Algorithm 1 to determine $J$ and $J_p$ from the data. The optimization for both experimental paths was performed jointly, and therefore, the fit parameters are the same for both cases. Our approach was found to yield excellent predictions, both qualitatively and quantitatively, relative to the results of a detailed and expensive analytical fit, as shown in Fig. 3a, b. The analytical parameters in the LSTW limit, adapted from Petsch et al.[28], are $J = 29.00(8)$ meV and $J_p = 1.67(5)$ meV. The parameters obtained from our machine learning fitting are $J = 29.68$ meV and $J_p = 1.70$ meV. We also experimented with fitting each path

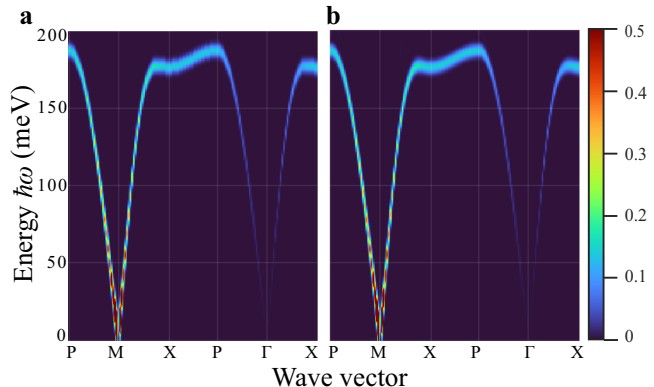

**Fig. 2 | Comparison between linear spin wave theory simulation and machine learning prediction for a given set of parameter values ($J = 45.57$ meV and $J_p = 2.45$ meV). a** Example of ground-truth simulated $S(\mathbf{Q}, \omega)$ calculated using the SpinW software program and **b** corresponding machine learning forward model prediction.

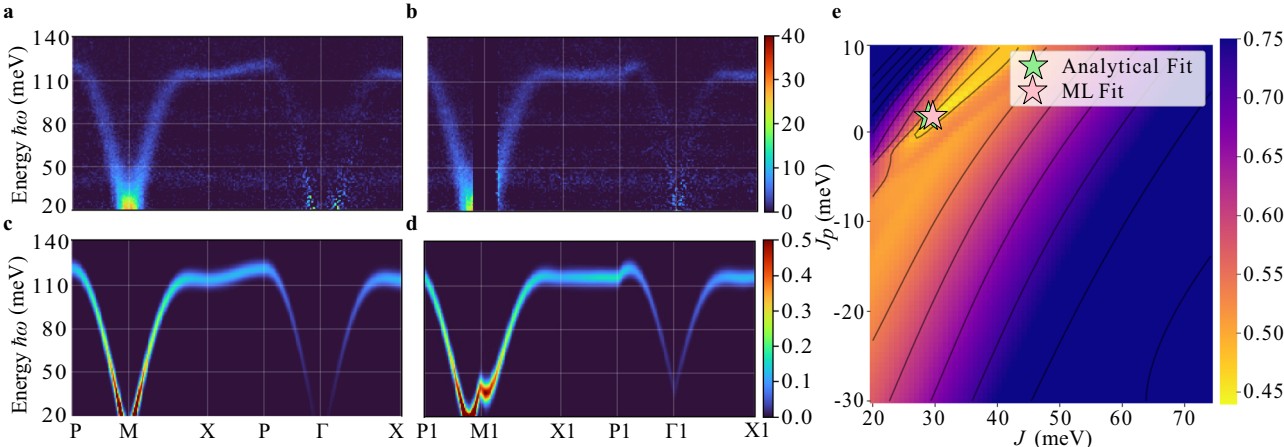

**Fig. 3 | Hamiltonian parameter extraction via auto-differentiation of the neural implicit representation. a, b** show experimental data after automated background subtraction. For the experimental data, the color bars reflect $S(\mathbf{Q}, \omega)$ in units of: mbarnsr$^{-1}$meV$^{-1}$f.u.$^{-1}$. **c, d** show the corresponding machine learning predictions for

the two paths. The predicted profiles are visually seen to be similar to the experimental data. Deviations at low $\hbar\omega$ are due to the neglect of anisotropic spin gaps in our model. **e** Visualization of the loss landscape for objective fitting in the Hamiltonian parameter space ($J, J_p$).

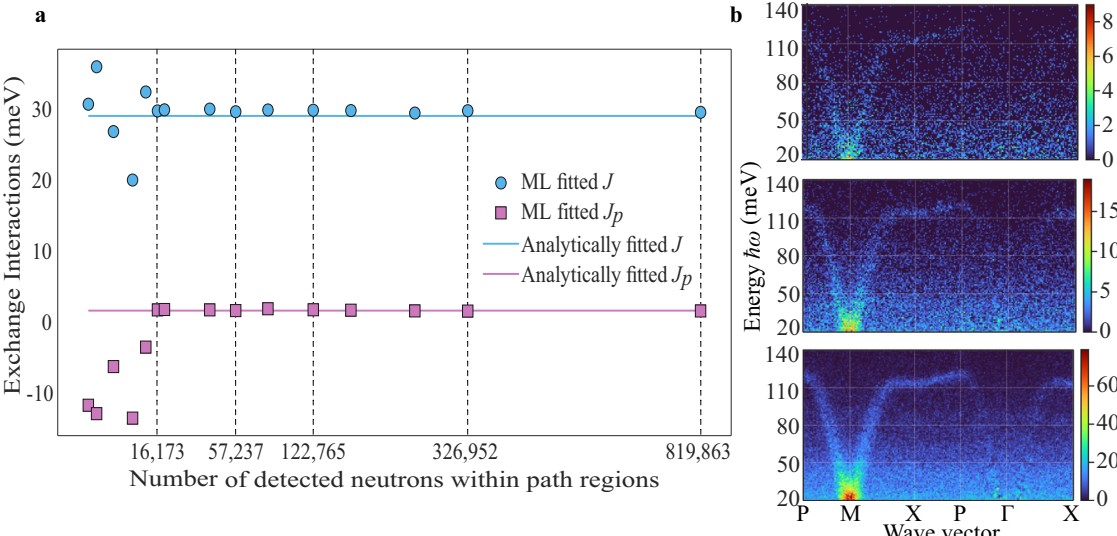

**Fig. 4 | Real-time Hamiltonian parameter estimation using a differentiable implicit neural representation. a** Machine learning prediction for $J$ and $J_p$ as a function of the total number of neutrons detected within the two path regions. Square-root scaling is used for the neutron counts due to the Poisson collection statistics. The machine learning prediction is seen to converge much earlier than the count-time recorded in the experiment. **b** Visualization of plausible low-count (without algorithmic background subtraction) data. Total number of neutrons detected with the two path regions are: 16,173, 57,237, and 326,952 (top to bottom). The colorbars show the absolute counts of detected neutrons.

independently and also obtained similar predictions; for path 2, this is especially notable since a significant portion of the experimental data is missing in this case, see Supplementary Fig. 3. Supplementary Figure 4 provides fitting results from SpinW with algorithmic peak-fitting, which yields similar results for this dataset.

Since our neural implicit model is computationally inexpensive to evaluate, we also constructed a loss landscape of the objective function with respect to $J$ and $J_p$. The objective function is found to be well-behaved and the gradient descent scheme finds a fit close to the analytical result (Fig. 3e). We emphasize that the only information provided to the algorithm is the knowledge of a region of the $(\mathbf{Q}, \hbar\omega)$-space on which to carry out an automatic background subtraction prior to fitting the data. Importantly, no peak finding or extraction is needed as the optimization objective uses the intensity of all provided voxels in the $(\mathbf{Q}, \hbar\omega)$-space or pixels on the 2D intensity map rather than the magnon peak positions $\hbar\omega_{\mathbf{Q}}$.

### Real-time fitting
In real experimental settings, another critical issue is the ability to make rapid decisions on whether or not sufficient data have been collected at any one time to allow for a good understanding of the physics being explored.

To probe the effectiveness of our framework for real-time fitting during an experiment, and to reduce data collection time, we used the experimental data to generate plausible data for low counting situations. Specifically, we smoothed the experimental data and used it as a probability distribution which is sampled using rejection sampling, see Methods. In a real experiment, a sample is normally measured using a series of different orientations on the spectrometer, often with varying time scales. Here, the rejection sampling simulates the La$_2$NiO$_4$ neutron scattering experiment performed in the same sample orientations but with shorter data collection times. This exercise gives insight into the viability of the approach for handling low statistics and noisy data. We note that any "detector noise" and scattering from the sample environment is negligible compared to statistical noise in the scattering from the sample. In Fig. 4a, we show the obtained parameters from the machine learning fitting as a function of the number of detected neutrons within the two path regions. Visualizations of path 1 at selected points in time are also shown in Fig. 4b. The machine learning

prediction is obtained as the lowest objective value from 10 independent gradient descent optimizations starting from random locations in Hamiltonian parameter space. Using the median prediction gives very similar results. This test demonstrates that our machine learning model quickly converges to the true solution and is effective under low signal-to-noise conditions.

## Discussion
In this work, we develop a neural implicit representation customized for inelastic neutron scattering analysis and show that this model can enable precise extraction of Hamiltonian parameters and has the potential to be deployed in real-time settings to minimize required counting time.

We emphasize that our implicit modeling scheme considers data as coordinates $(\mathbf{Q}, \omega, J, J_p)$ which is fundamentally different from the traditional image-based representations. One benefit of this approach is that the model continuously represents energy, momentum, and Hamiltonian parameters, and can therefore be used to make predictions at displaced coordinates $(\mathbf{Q} + \delta\mathbf{Q}, \omega + \delta\omega, J + \delta J, J_p + \delta J_p)$. This enables prediction at finer resolutions of $\mathbf{Q}$ and $\omega$ than those recorded on pixelized detectors or at Hamiltonian parameters not present in the training set. Additionally, since the model is a SIREN neural network, it is composed of a series of differentiable operations and is therefore amenable to automatic differentiation techniques. This is highly advantageous and allows the entire analysis pipeline to be compactly expressed by a single model that is end-to-end differentiable relative to the parameters of interest. This approach also allows for an elegant treatment of missing data. Here, missing coordinates can simply be dropped from the parameter estimation step without the need for additional model retraining or data masking.

To validate our approach, we use inelastic scattering data from La$_2$NiO$_4$ and find that our method accurately recovers unknown parameters corresponding to the assumed spin-1 Heisenberg Hamiltonian model on a square lattice. The small overestimation of $J$ arises from several factors. Small differences in the value of $J$ arise from the 3-dimensionality of $\mathbf{Q}$ and the associated variations in the magnetic form factors and polarization factors. Such 3-dimensional information is not included in our analysis since we only consider quantities averaged over $Q_z \in [-10, 10]$ r.l.u. Also, the resolution function and finite

lifetime are only approximations here and further, any multi-magnon scattering is not described by LSWT. Finally, we do not include effects of the experimentally observed energy shifts resulting from the spin gaps[28,34]. These issues could, however, be addressed through more comprehensive simulations within the overall modeling framework presented here.

Another area for improvement concerns the challenging task of background subtraction. For the analysis of $La_2NiO_4$, we were able to develop an automatic background subtraction scheme, based on human insight, to successfully suppress non-magnetic contributions which include non-magnetic coherent excitations (phonons here). However, the suppression of other contributions by this method may not always be feasible. In future work - phonon dispersion calculations, nuclear structure factors, and usage of **Q**-dependence of spectral weights - could be implemented in our framework to distinguish additional coherent excitations.

Our ability to continuously fit and refine data as it is collected is important for enabling more efficient and informative experimental design. Since neutron scattering measurements typically involve low detector count rates, this is a major factor that will influence the efficiency of measurement time at facilities. Moreover, one would like to minimize the amount of time needed to complete an experiment without sacrificing data quality. We have shown our model to perform well under low signal conditions and to yield accurate Hamiltonian parameter predictions, thereby providing guidance on when best to conclude data collection. Here, stochastic gradient descent of the neural implicit model is an effective strategy to filter noise and achieve robust optimization. Note that, if other paths in reciprocal space were available, leveraging the information obtained in the additional data would have simply required training with additional simulations, without any necessary changes to the overall machine-learning model. This is an important point for real-time applications, as the flexibility of the coordinate-based representation to ingest additional data is a significant advantage over from conventional analysis pipelines, which rely on manually guided peak-fitting algorithms that are not suited to this type of high-dimensional data. We note that the characterization of the framework's effectiveness for real-time fitting only considers the case of shorter counting times across all measured sample orientations, highlighting the framework's capability to handle sparsely distributed detection. Since such measurements usually have to be repeated, this analysis approach could be applied between repetitions to determine whether more data collection is necessary. Furthermore, additional work could involve simulating the training data with respect to sample orientations, which would be preferred when considering experimental guiding for a real, live experiment. In general, we anticipate that our method will be readily compatible with autonomous experimental steering agents by exploiting the model's fast and scalable forward computations which are essential in Bayesian experimental design[35,36].

Although the present contribution focused on linear spin wave simulations, the approach presented here is not restricted to a particular choice of theoretical scheme. We expect that our framework will be particularly impactful when combined with using expensive and advanced computational methods for simulating strongly correlated systems, such as exact diagonalization (ED)[37], density matrix renormalization group (DMRG)[38,39], determinant quantum Monte-Carlo (DQMC)[40,41], and variational Monte Carlo (VMC)[42,43] simulations.

The methodology presented here breaks the barrier of real-time fitting of inelastic neutron and x-ray scattering data, bypassing the need for complex peak-fitting algorithms or user-intensive post-processing. Our study thus opens new opportunities for significantly improved analysis of excitations in classical and quantum systems.

## Methods

### Sample preparation and data collection

In the experiment, a 21 g single crystal of the quasi-2D Néel antiferromagnet $La_2NiO_{4+\delta}$ ($P4_2/ncm$ with $a = b = 5.50$ Å and $c = 12.55$ Å), grown by the floating-zone technique, was utilized. The presented time-of-flight neutron spectroscopy data were collected on the SEQUOIA instrument at the Spallation Neutron Source at the Oak Ridge National Laboratory[44] with an incident neutron energy of 190 meV, the high-flux Fermi chopper spun at 300 Hz, and a sample temperature of 6 K. The data is integrated over the out-of-plane momentum $Q_z \in \pm 10$ r.l.u. The lattice can be approximated by $I4/mmm$ with $a = b \approx 3.89$ Å. $Q_x$ and $Q_y$ for $I4/mmm$ are equivalent to $Q_x$ and $Q_y$ in the square-lattice notation. For more details see ref. 28.

### SpinW simulation and fitting

In an inelastic scattering experiment, the measured quantity is the partial differential cross section which is related to the dynamical structure factor $S(\mathbf{Q}, \omega)$ by $\frac{d^2\sigma}{d\Omega dE_f} = k_f/k_i \, S(\mathbf{Q}, \omega)$, where $k_i$ and $k_f$ are the incident and final neutron or photon wave vectors. In our simulations, the dynamical structure factor is approximated to $S(\mathbf{Q}, \omega) \propto \sum_{m,n} \int dt \, e^{-i\mathbf{Q}\cdot(\mathbf{r}_m - \mathbf{r}_n)} e^{-i\omega t} \langle S_m(t) S_n(0) \rangle$, where $\langle S_m(t) S_n(0) \rangle$ represents spin-spin correlations at different atomic sites $m, n$. The neutron polarization factor as well as the magnetic form factor are neglected here.

The two momentum paths used for $S(\mathbf{Q}, \omega)$ simulation are $\mathbf{Q}_{list1} = \left\{ \left[\frac{3}{4} \frac{1}{4} 0\right], \left[\frac{1}{2} \frac{1}{2} 0\right], \left[\frac{1}{2} 0 0\right], \left[\frac{3}{4} \frac{1}{4} 0\right], [1 0 0], \left[\frac{1}{2} 0 0\right] \right\}$ and $\mathbf{Q}_{list2} = [-0.07 \, 0.03 \, 0] + \mathbf{Q}_{list1}$, respectively in reciprocal lattice units. The SpinW software[12] was used to perform 600 simulations for each path (1200 total) corresponding to randomly sampling $J$ and $J_p$ in ranges of [20, 75] meV and [-30, 10] meV. The lower limit for $J$ and upper limit for $J_p$ are chosen such that the ground state remains the Néel state which is satisfied in LSWT for $J > 2J_p$ and $J > 0$. For each location in **Q**, the corresponding energies from 0 to 200 meV were obtained. The quantum fluctuation renormalization factor $Z_c$ is set to 1.09[28,45,46]. After simulation, the data was convoluted with an energy-dependent kernel based on the beamline instrument profile. For this procedure, an in-built tool from SEQUOIA was used to give a polynomial fit for the dependence of the resolution (FWHM) in meV on the energy transfer ($\hbar\omega$) in meV: FWHM $= 1.4858 \times 10^{-7}(\hbar\omega)^3 + 1.2873 \times 10^{-4}(\hbar\omega)^2 - 0.084492\hbar\omega + 14.324$[44]. In addition, the data was broadened with a 1D Gaussian kernel ($\sigma = 5$ pixels) in **Q** to correct for the discrete sampling of the simulation and to partially consider the momentum resolution of the instrument.

The SpinW-software-based spin wave spectrum fitting was implemented using its built-in function. The inputs are peak information extracted from experimental spin wave dispersion data. The $R$ value is optimized using a particle swarm algorithm to find the global minimum defined as $R = \sqrt{1/n_E \times \sum_{i,\mathbf{Q}} 1/\sigma_{i,q}^2 (\hbar\omega_{i,\mathbf{Q}}^{sim} - \hbar\omega_{i,q}^{meas})^2}$, where $(i, q)$ index the spin wave mode and momentum, respectively. $E_{sim}$ and $E_{meas}$ are the simulated and measured spin wave energies, $\sigma$ is the standard deviation of the measured spin wave energy determined previously by fitting the inelastic peak and $n_E$ is the number of energies to fit.

### SIREN model training

A 5-layer SIREN neural network (Fig. 1c) was trained on 1000 simulations of $(S(\mathbf{Q}, \omega), J, J_p)$ tuples; 200 simulations were left aside for validation and testing. Here, $\hbar\omega \in [0-200]$ meV, $J \in [20 - 75]$ meV and $J_p \in [-30 - 10]$ meV were normalized to 0-1 in order for all the parameters to be on approximately the same scale. The model was trained to predict $\log(1 + S(\mathbf{Q}, \omega, J, J_p))$ by optimizing the mean-squared-error objective $L$ between the prediction and the label with respect to the network parameters. During training, the following hyperparameters and settings were used: Adaptive Moment Estimation (ADAM)

algorithm for optimization ($\beta_1 = 0.9$, $\beta_2 = 0.999$)[31], batch size = 2048, learning rate = 0.001. Here, $\beta_1$ and $\beta_2$ influence the degree to which past gradients affect the current step. The batch size is a parameter that controls the number of images used to compute the mean-squared-error objective and the learning rate controls the gradient descent step size. The learning rate was exponentially decayed by a factor of $\exp(-0.1)$ for every epoch (full pass through the entire dataset) after the first ten epochs. We used NVIDIA A100 GPU hardware with the Keras API[47] and the model was trained for 50 epochs.

**Machine learning parameter extraction**
Prior to differentiable optimization, the experimental data were automatically background subtracted using the following procedure. First, a region of ($\mathbf{Q}_{list}$, $\omega_{list}$) space was chosen for each slice (160-170 pixel location in the $\mathbf{Q}$-axis) and averaged across $\mathbf{Q}_{list}$ to yield a one-dimensional energy profiles. This procedure was chosen based on prior assumptions on the isotropic nature of the scattering and the Néel ground state. Next, the one-dimensional energy profiles were fit using a Savitzy-Golay filter (window size = 51, polynomial order = 3) and used for background subtraction.

The unknown $J$ and $J_p$ parameters were recovered from data using gradient-based optimization of the neural network implicit representation. For the experimental data presented in this work, the metric $(1 - r)$ between the measured and simulated $(1 + S(\mathbf{Q}, \omega, J, J_p))$ was used as the objective function $L$ introduced in Equation (3); here, $r$ refers to the Pearson correlation coefficient. No normalization was performed for scaling the simulation data relative to the experimental data.

The objective $L$ was optimized using the ADAM algorithm with respect to $J$ and $J_p$ and $\mathbf{Q}_{list}$ and $\omega_{list}$ were randomly sampled from the list of paths containing the experimental data. Here, a batch size of 4096 was used for the ($\mathbf{Q}_{list}$, $\omega_{list}$) sampling, with 2000 Adam optimization steps and a learning rate of 0.005. Here, the batch size refers to the number of pixels in the experimental image that are randomly selected in each step of the optimization procedure.

**Low count data generation and fitting**
High-count data for each slice (without background subtraction) were smoothed using a $3 \times 3$ Gaussian convolutional kernel. The resultant images were each normalized to (0, 1) using the total intensity. Each slice was treated as a probability distribution which was sampled using Monte-Carlo rejection sampling. This process was used to create a series of datasets with neutron counts in the range ($1 \times 10^4 - 9 \times 10^6$). Each dataset was individually and automatically background subtracted by the previously described method and fit ten times from random starting locations in ($J$, $J_p$) using the machine learning optimization procedure. Note, the corresponding low-count data was used in order to perform the automated background subtraction.

## Data availability
All data generated in this study as well as a minimal dataset have been deposited in the Zenodo database under accession code https://doi.org/10.5281/zenodo.8267499[48].

## Code availability
The code developed in this study have been deposited in the Zenodo database under accession code https://doi.org/10.5281/zenodo.8267474[49] and is also available at https://github.com/slaclab/neural-representation-sqw.git.

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

## Acknowledgements

This work is supported by the U.S. Department of Energy, Office of Science, Basic Energy Sciences under Award No. DE-SC0022216, as well as under Contract DE-AC02-76SF00515, both for the Materials Sciences and Engineering Division, as well as for the Linac Coherent Light Source (LCLS), part of the Scientific User Facilities Division. A portion of this research used resources at the Spallation Neutron Source, a DOE Office of Science User Facility operated by the Oak Ridge National Laboratory. J.J. Turner acknowledges support from the U.S. DOE, Office of Science, Basic Energy Sciences through the Early Career Research Program. Z.J. is supported by the Stanford Science fellowship, and the Urbanek-Chodorow postdoctoral fellowship awards. A.N. Petsch and S.M. Hayden acknowledge funding and support from the Engineering and Physical Sciences Research Council (EPSRC) under Grant Nos. EP/L015544/1 and EP/R011141/1.

## Author contributions

S.R.C., Z.J., and A.N.P. contributed equally to this work and focused on the machine learning, simulation, and experimental portions respectively. A.I.K. and S.M.H. assisted with data collection. C.P., Z.C., R.P., H.C., S.M., M.D., S.C., A.B., A.F., D.P., and D.R. assisted with data analysis and manuscript writing. C.J., Y.N., and J.T. supervised the work.

## Competing interests

The authors declare no competing interests.
