## [Peer Review File · Nature Communications]

REVIEWER COMMENTS

Reviewer #1 (Remarks to the Author):

In this work the authors describe their development of an artificial intelligence framework to determine unknown Hamiltonian parameters from inelastic neutron scattering experiments. Their framework combines a neural network, previously trained on simulated data from a model Heisenberg Hamiltonian, with automatic differentiation. This approach is successfully tested on a Linear Spin Wave Theory (LSWT) simulator and experimental data from the square-lattice spin-1 antiferromagnet La₂NiO₄.

While this work provides some new results worthy of publication, as a whole it is not of sufficient novelty to warrant publication in this journal. By the authors own admission, this work extend recent ML guided analysis of static neutron scattering experiments (such as refs 16 and 17). This would be already a concern when judging novelty and, moreover, there is already prior work in the literature that tackles the problem of inelastic neutron scattering: in addition to references 18 and 19, which train inverse models on inelastic neutron scattering data, one can find that [Doucet, Mathieu, et al. "Machine learning for neutron scattering at ORNL." Machine Learning: Science and Technology 2.2 (2020): 023001.] gives an overview that includes references to inelastic neutron scattering and [Samarakoon, Anjana M., et al. "Extraction of interaction parameters for α -RuCl₃ from neutron data using machine learning." Physical Review Research 4.2 (2022): L022061.] performs a full analysis on inelastic neutron scattering results using ML methods.

This is not to say that the contribution of this work is irrelevant, quite the contrary, it introduces novel aspects to the problem that are worthy of publication, but this should be done in a more specialised journal.

Minor corrections

- Line 86: it should say conventional FFNN, not convention FFNN.

- Line 138: SIREN should be defined on first use.

- Some passages become a bit repetitive: compare for example lines 151- 153 and eq 2 or lines 159-160 and lines 179-180

- Line 167: Since this article is probably intended for a wider readership it would be good to explicitly label all referenced BZ critical points in Fig 1b (P is missing).

Reviewer #2 (Remarks to the Author):

The authors present a new method for applying neural-network techniques to the modeling of measured excitations in a crystal. Here they apply the approach to determining superexchange parameters for spin waves in antiferromagnetic La_2NiO_4 , making use of recent experimental results obtained by time-of-flight neutron scattering. The method involves training the neural network on model calculations for a range of parameter values and then finding the result that gives the best match to the experimental data. If applied in real time during an experiment, this approach has the potential to optimize measurement time.

In my opinion, this new approach to data analysis is quite interesting. I would support acceptance after the authors have had a chance to consider the following comments.

1. Please insert each equation between lines within the paragraph where it is discussed, not at the end of the paragraph.

2. Some notation is used that may be common among neural network experts but which is unfamiliar to an experimentalist such as myself. Please present more definitions. For example, please explain the set theory notation first used in line 152.

3. Please give a better explanation for Φ . Equation (2) is not an equation: what does the right arrow mean? Also, what is the connection to Fig. 1(c), which is supposed to provide a visualization. Related to that, the figure shows "64 x 3" twice, while the caption mentions 64 x 3 and 64 x 1; why?

4. The potential application to real-time analysis during data collection seems to assume that the full Q-dependence is collected simultaneously. I suspect (based on my own experience) that the experimental data shown required measurements at many different sample angles. Of course, such measurements can be repeated, and the analysis approach could be applied before repetitions to determine whether they are necessary. Noting this sort of reality would be appropriate.

5. The real-time analysis proposal also neglects possible complications from features such as phonons. In the present case, it appears that the “background” subtraction scheme, along with the large L-integration range, has been effective in minimizing phonon signal relative to spin waves. This may not always be the case (or at least an effective way to minimize the phonon signal might not be obvious in real time).

Reviewer #3 (Remarks to the Author):

The authors report on a developed AI framework combining a neural network trained to mime simulated data from a model Hamiltonian with automatic differentiation to recover unknown parameters from experimental data, and benchmark the approach on a LSWT simulator and an experimental INS dataset.

This work meets the current challenges to use neutron and photon beamtime as efficient as possible, and to make use of advanced computing opportunities on the other hand. The strength of the presented paper lies in a thorough analysis of the existing work in the field, the focus on a well-defined problem complementary to existing approaches (predicting continuous parameters within an assumed Heisenberg model during neutron/x-ray experiments), and considering the specific aspects of the physics and experimental technique (rapid decisions on sufficient statistics, weighting weak features in the spectrum, background consideration, instrument resolution, low neutron experiment count-rates). The clear definition of the problem to answer and the limitation of methodological development on that allows to present an approach which identifies successfully the key parameters of a linear spin-wave spectrum.

The work is justified in detail. The paper is very well structured and explained, it allows both specialists in solid-state physics and AI methods to follow a well-thought strategy, its different steps and the reasons to choose the methods and to neglect aspects in this state of the work.

Being not a specialist in AI methods the methodological decisions are reasonable to me. The benchmark on La₂NiO₄ is carefully performed and provides very good results with respect to physics and ML method. The idea to create a surrogate model in advance making an in-situ parameter estimate during the experiment possible is excellent. Providing suited tools in addition could be a big step in changing certain inelastic experiment procedures to be more efficient.

I therefore suggest to publish the work giving huge impact to the experimental method.

There are only few minor comments:

- Fig. 1c: May there be a typo in the given layers? Two times 64×3 instead of 64×1 and 64×3 ? In case the figure is right, please explain.
- Fig.2: For sure the colour plots look very similar but the quantitative comparison is difficult by eyes. Would it be reasonable to show the subtraction or any other quantitative criteria for the agreement?
- Line 338: number of detected neutrons within the path region – could this be so clearly stated in the caption of Fig. 4, too, instead of “Detector Counts”?

Reviewer #1 (Remarks to the Author):

Reviewer Comments 1.1: *In this work the authors describe their development of an artificial intelligence framework to determine unknown Hamiltonian parameters from inelastic neutron scattering experiments. Their framework combines a neural network, previously trained on simulated data from a model Heisenberg Hamiltonian, with automatic differentiation. This approach is successfully tested on a Linear Spin Wave Theory (LSWT) simulator and experimental data from the square-lattice spin-1 antiferromagnet La₂NiO₄.*

*While this work provides some new results worthy of publication, as a whole it is not of sufficient novelty to warrant publication in this journal. By the authors own admission, this work extends recent ML guided analysis of static neutron scattering experiments (such as refs 16 and 17). This would be already a concern when judging novelty and, moreover, there is already prior work in the literature that tackles the problem of inelastic neutron scattering: in addition to references 18 and 19, which train inverse models on inelastic neutron scattering data, one can find that [Doucet, Mathieu, et al. "Machine learning for neutron scattering at ORNL." *Machine Learning: Science and Technology* 2.2 (2020): 023001.] gives an overview that includes references to inelastic neutron scattering and [Samarakoon, Anjana M., et al. "Extraction of interaction parameters for α -RuCl₃ from neutron data using machine learning." *Physical Review Research* 4.2 (2022): L022061.] performs a full analysis on inelastic neutron scattering results using ML methods.*

This is not to say that the contribution of this work is irrelevant, quite the contrary, it introduces novel aspects to the problem that are worthy of publication, but this should be done in a more specialised journal.

Our Response 1.1: We appreciate your comments regarding the technical quality of our work and your other thoughtful remarks. Here, we take the opportunity to point out some of the novel aspects of the work that we did not properly emphasize in our original submission.

In short, the novelty of our study involves three major aspects that will be of interest to various scientific communities:

- Use of implicit neural representations as a new way of implementing ML in experimental work.
- Real-time data interpretation of our model, which is key for future efforts towards experimental guiding and optimization.
- Simplicity of our model. Our straightforward platform will be amenable to implementation by non-experts and it will thus be adaptable to many areas beyond neutron scattering.

We elaborate further on the preceding points as follows.

- We incorrectly noted in our original manuscript that our work “extends” prior ML efforts for neutron scattering data. We emphasize that our work does not simply build on or extend prior work but that our study presents a fundamentally new approach. Although References 1-3 use a forward model for Hamiltonian parameter extraction, our approach is very different. Our novelty is in the application of implicit neural representations as a powerful tool to model scattering experiments. We show that this approach, which has recently been shown to accurately represent images and hypervolumes in computational photography, can add substantial value to analyzing dynamical measurements.
- Our framework is based on a coordinate representation of the data as opposed to the pixelated representations used in all prior works, and therefore, it does not rely on grid discretization. This implies that our trained forward model can be applied to essentially arbitrary ranges of the energy/momentum space that are provided by various detectors and instrument configurations. In addition, our approach can take in unstructured data and is not sensitive to input sizes. For example, it does not require data to be formatted or padded into 2D/3D grids.
- Our approach yields a simple and compact solution via an architecture that allows the entire pipeline to be described by one differentiable model rather than a combination of models which have to be separately optimized and trained. Furthermore, our coordinate representation is able to naturally handle cases such as missing or additional data which may pose issues for pixel-based representation methods.
- Another major point of distinction between our method and that of Refs. 2 and 3 is that our focus is on being able to continuously fit parameters in real-time as data are collected. Refs. 2 and 3 instead focus on selecting which simulations to perform in an intelligent fashion. We show that automatic differentiation via stochastic gradient descent is effective at filtering noise and that our approach performs well under low-signal conditions typical of real-time measurements. We believe that the continuous fitting capability of our approach will be valuable in determining when sufficient data has been collected to terminate an experiment as well as in guiding where the measurement should next be performed.

[1] Samarakoon, Anjana M., et al. "Machine-learning-assisted insight into spin ice Dy₂Ti₂O₇." *Nature communications* 11.1 (2020): 892.

[2] Samarakoon, Anjana, et al. "Integration of machine learning with neutron scattering for the Hamiltonian tuning of spin ice under pressure." *Communications Materials* 3.1 (2022): 84.

[3] Samarakoon, Anjana M., et al. "Extraction of interaction parameters for α -RuCl₃ from neutron data using machine learning." *Physical Review Research* 4.2 (2022): L022061.

[4] Doucet, Mathieu, et al. "Machine learning for neutron scattering at ORNL." *Machine Learning: Science and Technology* 2.2 (2020): 023001.

Revisions made in response to Comments 1.1. The preceding discussion has been incorporated briefly in the revised manuscript as follows.

- Refs. 3 and 4 are included as you suggested. A paragraph has been consolidated into the Introduction section (Lines 62-85) which describes previous ML work in further detail.
- We have moved the explanation of the neural implicit representation idea to the Introduction (Lines 86-104) to further emphasize the novelty of applying this method to inelastic neutron scattering.
- Further text has been added in the discussion section on the ability of our approach to readily ingest new data, handle missing data without the need for model retraining, and predicting under low signal-to-noise conditions (Lines 304-327, 363-376).

Reviewer Comments and our Responses 1.2: *Minor corrections*

- *Line 86: it should say conventional FFNN, not convention FFNN.*

Thank you for the correction, we noticed that the model used for this task was actually a convolutional neural network (instead of a FFNN). We have made this correction in the text (Lines 65-68).

- *Line 138: SIREN should be defined on first use.*

Thank you for the clarifying comment. We have added the description Sinusoidal Representation Network (SIREN) to the text (Line 142).

- *Some passages become a bit repetitive: compare for example lines 151- 153 and eq 2 or lines 159-160 and lines 179-180*

We have removed Equation 2 and replaced it with a mathematical definition of the SIREN function (ending Line 163) in response to a comment from Reviewer 2.

The description of the motivation for the $\log(1+x)$ transform now only appears once in the manuscript (Lines 152-154).

- *Line 167: Since this article is probably intended for a wider readership it would be good to explicitly label all referenced BZ critical points in Fig 1b (P is missing).*

Thank you for this comment. We have added the missing point to the revised Figure 1b.

Reviewer #2 (Remarks to the Author):

Reviewer Comments 2.1: *The authors present a new method for applying neural-network techniques to the modeling of measured excitations in a crystal. Here they apply the approach to determining superexchange parameters for spin waves in antiferromagnetic La_2NiO_4 , making use of recent experimental results obtained by time-of-flight neutron scattering. The method involves training the neural network on model calculations for a range of parameter values and then finding the result that gives the best match to the experimental data. If applied in real time during an experiment, this approach has the potential to optimize measurement time.*

In my opinion, this new approach to data analysis is quite interesting. I would support acceptance after the authors have had a chance to consider the following comments.

Our Response 2.1: Thank you very much for your review. We found your questions very helpful and hope that the amendments will strengthen the work and make it more accessible to the readership. Furthermore, we have included your suggestions about highlighting relevant experimental conditions which may pose issues for the ML methodology presented here.

Reviewer Comments and our Responses 2.2:

1. Please insert each equation between lines within the paragraph where it is discussed, not at the end of the paragraph.

Thank you for this suggestion, which is now followed in the revised manuscript.

2. Some notation is used that may be common among neural network experts but which is unfamiliar to an experimentalist such as myself. Please present more definitions. For example, please explain the set theory notation first used in line 152.

We are happy to follow your thoughtful suggestion. We have included a few changes to better present the work to experimentalists:

- We have now added definition for R_{+}^1 being a real positive number $\{x \in \mathbb{R} | x > 0\}$ (Line 146)
- We have added details of SIREN as an acronym for Sinusoidal Representation Network (Line 142).
- We have added a brief explanation of the key parameters that govern ML training in the Methods: SIREN Model Training section (Lines 480-484).
- We have added a brief explanation for the batch size parameter for the auto-differentiation of the ML model in the Methods: Machine Learning Parameter Extraction section (Lines 513-515)

3. Please give a better explanation for Φ . Equation (2) is not an equation: what does the right arrow mean? Also, what is the connection to Fig. 1(c), which is supposed to provide a visualization. Related to that, the figure shows “64 x 3” twice, while the caption mentions 64 x 3 and 64 x 1, why? Please give a better explanation for Φ . Equation (2) is not an equation: what does the right arrow mean?

Thank you for this question. Φ is the neural network which aims to approximate the function $\log(1+S(Q,w,J,J_p))$. Equation 2 was intended to represent this mapping, but we recognize that this is unclear and that it is not actually an equality. We have removed this equation from the text and instead provided a new Equation 2 which is the mathematical definition for the SIREN function. Note that the implicit neural representation is composed of a series of matrix multiplications and vector additions which are non-linearly transformed by successive applications of the sine function. The matrices and vectors used are “learned” during the optimization procedure in order to allow Φ to closely mimic $\log(1+S(Q,w,J,J_p))$. The preceding points are now clarified in the text when discussing Eq. 2 (Lines 154-163).

Also, what is the connection to Fig. 1(c), which is supposed to provide a visualization.

With the new Equation 2, Figure 1c now corresponds directly. It is the graphical representation of the series of matrix multiplications given by Equation 2.

Related to that, the figure shows “64 x 3” twice, while the caption mentions 64 x 3 and 64 x 1, why?

Thank you for this correction – this was a typo in the Figure. The figure has been amended to show 64x3 for the neural network layers with the sinusoidal activation function and 64x1 for the layer with the linear activation function. This is also clarified in the Figure 1c caption.

4. The potential application to real-time analysis during data collection seems to assume that the full Q -dependence is collected simultaneously. I suspect (based on my own experience) that the experimental data shown required measurements at many different sample angles. Of course, such measurements can be repeated, and the analysis approach could be applied before repetitions to determine whether they are necessary. Noting this sort of reality would be appropriate.

Thank you for this comment. We agree that the way in which the effectiveness of the framework for real-time analysis is examined assumes that the full Q -dependence has been collected. This exercise gives insight into the ability of the model to handle low statistics and noisy data. We added your comment in the text, that once a full data set has been collected, this could be used to determine if more data is needed. We also note that future work could involve simulating the training data with respect to sample orientations, which would be preferred when considering

experimental guiding for a real, live experiment. We adapted the manuscript to include these points in the section on Real-Time Fitting (Lines 282-289) in the Discussion (Lines 378-388).

5. The real-time analysis proposal also neglects possible complications from features such as phonons. In the present case, it appears that the “background” subtraction scheme, along with the large L-integration range, has been effective in minimizing phonon signal relative to spin waves. This may not always be the case (or at least an effective way to minimize the phonon signal might not be obvious in real time).

We agree that the specific “background” subtraction scheme applied here suppresses dispersionless phonon modes. This scheme is already challenging for strongly dispersive phonons or excitations of orbital nature. Implementing a framework for more complex background subtraction will be interesting for future work but it is beyond the scope of this work. We have added comments on this challenge in the Discussion section (Lines 345-355).

Reviewer #3 (Remarks to the Author):

Reviewer Comments 3.1: *The authors report on a developed AI framework combining a neural network trained to mime simulated data from a model Hamiltonian with automatic differentiation to recover unknown parameters from experimental data, and benchmark the approach on a LSWT simulator and an experimental INS dataset.*

This work meets the current challenges to use neutron and photon beamtime as efficient as possible, and to make use of advanced computing opportunities on the other hand. The strength of the presented paper lies in a thorough analysis of the existing work in the field, the focus on a well-defined problem complementary to existing approaches (predicting continuous parameters within an assumed Heisenberg model during neutron/x-ray experiments), and considering the specific aspects of the physics and experimental technique (rapid decisions on sufficient statistics, weighting weak features in the spectrum, background consideration, instrument resolution, low neutron experiment count-rates). The clear definition of the problem to answer and the limitation of methodological development on that allows to present an approach which identifies successfully the key parameters of a linear spin-wave spectrum.

The work is justified in detail. The paper is very well structured and explained, it allows both specialists in solid-state physics and AI methods to follow a well-thought strategy, its different steps and the reasons to choose the methods and to neglect aspects in this state of the work.

Being not a specialist in AI methods the methodological decisions are reasonable to me. The benchmark on La_2NiO_4 is carefully performed and provides very good results with respect to physics and ML method. The idea to create a surrogate model in advance making an in-situ parameter estimate during the experiment possible is excellent. Providing suited tools in addition could be a big step in changing certain inelastic experiment procedures to be more efficient.

I therefore suggest to publish the work giving huge impact to the experimental method.

Our Response 3.1: Thank you very much for your review. We appreciate the positive comments about the readability of the manuscript and the appropriateness of the approach to be applied to experimental settings. We include all your suggested changes which are detailed specifically below.

Reviewer Comments and our Responses 3.2:

There are only few minor comments:

•*Fig. 1c: May there be a typo in the given layers? Two times 64 x 3 instead of 64 x 1 and 64 x 3? In case the figure is right, please explain.*

Regarding Fig. 1c, this was indeed a typo. The figure has been amended to show 64x3 for the neural network layers with the sinusoidal activation function and 64x1 for the layer with the linear activation function. We have also clarified this point in the figure caption.

•*Fig.2: For sure the colour plots look very similar but the quantitative comparison is difficult by eyes. Would it be reasonable to show the subtraction or any other quantitative criteria for the agreement?*

Following your suggestion, we have now added an additional figure in the Supplementary Information section, which shows the absolute difference between the LSWT simulation and the ML prediction (Supplemental Figure S2). At the colorbar level which is used for the rest of the data in the manuscript (0.5), there are only small variations between the simulation and the ML prediction. Note, there appears to be some alternating patterns in the difference profile. This is likely due to the discrete choice of momenta simulated with SpinW. This is smoothed by the ML prediction which is continuous. The preceding points are noted in the caption to Figure S2.

•*Line 338: number of detected neutrons within the path region – could this be so clearly stated in the caption of Fig. 4, too, instead of “Detector Counts”?*

We have amended the Figure 4a x-axis label to “Number of detected neutrons within path regions” and corrected the caption appropriately.

REVIEWERS' COMMENTS

Reviewer #1 (Remarks to the Author):

The latest version of the manuscript exhibits significant improvements compared to its previous iteration. The authors have taken steps to address all comments and suggestions. In particular, the introduction now provides readers with a clearer and more comprehensive understanding of the research's scope and significance, and overall the new version adeptly highlights the novel aspects of the study, emphasizing the original contributions it brings to the field.

Although I retain some minor reservations regarding the novelty and potential impact of the research, I acknowledge that such assessments can be subjective. Considering the significant improvements made to the manuscript and taking into account the opinions of the other referees, I find no compelling reason to object to its publication.

Reviewer #2 (Remarks to the Author):

I have read the authors' responses to my comments and those of the other referees, as well as the changes to the manuscript. In my opinion, the authors have made reasonable modifications of the manuscript. I support acceptance of the revised manuscript.

Reviewer #3 (Remarks to the Author):

The authors report on an AI framework using a neural-network technique for modeling excitations from neutron (or x-ray) scattering data, and apply it on the example of spin waves in La_2NiO_4 measured by inelastic neutron scattering. The aim of the development is an in-situ determination of the model parameters (here: the exchange parameters) during the experiment.

The revised version of the paper is significantly improved in readability, especially for non-specialists (either in neutron scattering or in ML methods), and I appreciate the effort to consider the comments of the editor and the reviewers. The paper provides an (again improved) thorough analysis of the existing

work in the field including the specific aspects of the physics and experimental technique. The approach is well justified to be complementary to others and providing a new way for real-time interpretation of experimental data, by fitting model parameters with ML technique but without (human-guided) peak-fitting procedures.

Being not an expert in ML methods, I accept the novelty and the sense of the approach. Absolutely, the work contributes to the scientific discussion on the potential and the usefulness of ML techniques in neutron and x-ray scattering beyond image analysis, and will interest scientists to re-use the provided approach. This will lead to a discussion of the effort (e.g. number of simulations before the experiment) and benefit (e.g. beamtime saved) vs. the scientific outcome in the field of inelastic scattering, and drive the development of experimental procedures and possible cultural changes.

All reviewer comments have been considered.

Therefore, I recommend to publish the paper after considering the following point:

Thanks for preparing figure S2. I understand the alternating pattern in the difference profile. Nevertheless, I am not sure about the scale of the color code - if the values of $S(Q,w)$ in Fig.2 are mainly 0...0.5, the difference of both approaches (LSWT and ML prediction) cannot be of ± 0.4 in similar units. In addition, the difference plot should not mimic the shape of the dispersion, but should be more or less flat (especially since there are "there are only small variations between the simulation and the ML prediction" as mentioned in your reply. May be there is a different understanding of "difference plot"; please check again and either describe the difference plot in more detail or change the scale(s).

Response to Reviewers

We thank the reviewers for their thorough review of our manuscript and for their time and effort during this process. Having addressed the comments from peer review, we feel that the quality and impact of the manuscript has been substantially improved.

Below, we address the last remaining comment from Reviewer 3:

Thanks for preparing figure S2. I understand the alternating pattern in the difference profile. Nevertheless, I am not sure about the scale of the color code - if the values of $S(Q,w)$ in Fig.2 are mainly 0...0.5, the difference of both approaches (LSWT and ML prediction) cannot be of ± 0.4 in similar units. In addition, the difference plot should not mimic the shape of the dispersion, but should be more or less flat (especially since there are "there are only small variations between the simulation and the ML prediction" as mentioned in your reply. Maybe there is a different understanding of "difference plot"; please check again and either describe the difference plot in more detail or change the scale(s).

Thank you for your comment. The difference plot here is simply the prediction minus spinW simulation. We have changed the colorbar to reflect the maximal positive and negative deviation and updated the figure caption accordingly. In fact, we would expect that the difference plot might mimic the shape of the dispersion curve. Here, the difference is really highlighting the areas where the model may be uncertain. Note, in regions where the true signal is 0, the ML model is able to confidently predict that the value will be 0 (this is because neighboring pixels also have 0 values). However, there is higher uncertainty on the dispersion curve and so we would expect worse predictions in this region. This is particularly true since the simulated curve is discrete and the ML prediction is a continuous interpolation.

Capturing dynamical correlations using implicit neural representations

Sathya Chitturi,^{1,2*} Zhurun Ji,^{3,4*} Alexander N. Petsch,^{1,5,6*} Cheng Peng,⁵ Zhantao Chen,⁵ Rajan Plumley,^{1,5,7} Mike Dunne,¹ S. Mardanya,⁸ S. Chowdhury,⁸ H. Chen,⁹ A. Bansil,⁹ A. Feiguin,⁹ A. I. Kolesnikov,¹⁰ D. Prabhakaran,¹¹ S. M. Hayden,⁶ Daniel Ratner,¹ Chunjing Jia,^{1,5,12} Youssef Nashed,¹ and Joshua J. Turner^{1,5*}

¹*SLAC National Accelerator Laboratory, Menlo Park, CA 94025, USA*

²*Department of Materials Science and Engineering, Stanford University, Stanford, CA 94305, USA*

³*Department of Physics and Applied Physics, Stanford University, Stanford, CA 94305, USA*

⁴*Geballe Laboratory for Advanced Materials, Stanford University, Stanford, CA 94305, USA*

⁵*Stanford Institute for Materials and Energy Sciences, Stanford University, Stanford, CA 94305, USA*

⁶*H.H. Wills Physics Laboratory, University of Bristol, Bristol BS8 1TL, United Kingdom*

⁷*Department of Physics, Carnegie Mellon University, Pittsburgh, PA 15213, USA*

⁸*Department of Physics and Astrophysics, Howard University, Washington, USA*

⁹*Department of Physics, Northeastern University, Boston, USA*

¹⁰*Neutron Scattering Division, Oak Ridge National Laboratory, Oak Ridge, Tennessee 37831, USA*

¹¹*Department of Physics, University of Oxford, Clarendon Laboratory, Oxford OX1 3PU, United Kingdom*

¹²*Department of Physics, University of Florida, Gainesville, FL 32611, USA*

(Dated: August 22, 2023)

Understanding the nature and origin of collective excitations in materials is of fundamental importance for unraveling the underlying physics of a many-body system. Excitation spectra are usually obtained by measuring the dynamical structure factor, $S(\mathbf{Q}, \omega)$, using inelastic neutron or x-ray scattering techniques and are analyzed by comparing the experimental results against calculated predictions. We introduce a data-driven analysis tool which leverages ‘neural implicit representations’ that are specifically tailored for handling spectrographic measurements and are able to efficiently obtain unknown parameters from experimental data via automatic differentiation. In this work, we employ linear spin wave theory simulations to train a machine learning platform, enabling precise exchange parameter extraction from inelastic neutron scattering data on the square-lattice spin-1 antiferromagnet La_2NiO_4 , showcasing a viable pathway towards automatic refinement of advanced models for ordered magnetic systems.

INTRODUCTION

Quantum matter, as featured by the existence of macroscopic orders from microscopic spin and/or charge arrangements or other phases with spontaneous symmetry breaking, represents an abundant and complex class of materials in condensed matter physics. For example, the magnetic configuration of a material and its dynamics are often driven by competing effects of multiple interactions as well as crystalline symmetries. The collective spin excitations in most magnetic materials, such as spin waves or magnons, act as probes of those interactions. The associated dispersion relations and correlations are key for developing potential applications, which include next generation spintronics devices, as well as new strategies for carrying, transferring, and storing information [1–3].

A primary aim of the last few decades has been to characterize wide classes of excitations, and this has been facilitated by advances in spectroscopic techniques, such as neutron scattering [4–7]. These techniques use the kinematics of scattered neutrons to obtain dispersion relations, lifetimes, and amplitudes of spin excitations. Neutron scattering studies are, however, challenging due to the paucity of available neutron sources, low neutron flux compared to other sources, and small neutron scattering cross sections. As a result, the question of how the efficiency of neutron experiments could be enhanced is drawing considerable interest in the field [8, 9]. Notably, the interpretation of neutron scattering spectra can be

challenging and time-consuming due to the complex nature of the physical processes involved, the diversity of samples, and the limited knowledge often provided by theoretical modeling. It is clear that there is an urgent need for collaboration among experiment, theory, and data science to accelerate the understanding of spin-related properties of materials [10].

As the rates of data collection continue to increase rapidly, especially with the advent of next-generation X-ray free electron laser facilities and the ability to collect hyper-dimensional datasets, it is important to develop techniques for real-time modeling and analysis of experimental spectra. The ability to perform ‘on-the-fly’ fitting [11] would enable efficient use of expensive beamtime by ascertaining when sufficient data has been collected, as well as by coupling to adaptive sampling methods to gain the most information about parameters of interest with the least number of measurements. Currently, real-time fitting for neutron scattering data can require substantial preparation. For example, direct fitting with the software package SpinW [12] requires the extraction of the eigenmodes of the system and therefore, needs an accurate, and preferably automatic, peak extraction algorithm. When the chosen paths in reciprocal space are numerous or the dispersion relations change significantly along those paths, this can involve significant human guidance and monitoring. In addition, fitting directly with SpinW does not take into consideration the magnon peak intensities or their shapes. Approaches to fit peak intensities and shapes directly, such as Multi- or Tobyfit implemented in HORACE [13], are possible alternatives –

however, these fitting procedures still require significant human guidance and are either slow and therefore incompatible with data acquisition rates or else they require an analytical, rapidly calculable spin wave model. Finding such a model is usually only feasible for simpler systems with minimal magnetic frustration or a low number of magnetically distinct sites.

Machine learning methods have recently been utilized in the analysis of x-ray and neutron scattering measurements to improve the accuracy and speed of data interpretation [10, 14]. Convolutional neural networks, trained on linear spin wave simulations, have been applied to inelastic neutron scattering measurements to discriminate between two plausible magnetic exchange models [14, 15]. As these models are typically trained on simulated profiles, achieving robust prediction generally requires detailed modelling and corrective dataset augmentations of experimental effects accounting for attributes such as background noise, missing data, and matching instrumental profiles [15–17]. Recently, a cycleGAN approach, which makes experimental data look like simulated data has been proposed as a way to improve model robustness [16]. In cases where the desired observables have continuous values, such approaches are often highly sensitive to background noise and other effects [18]. To predict continuous Hamiltonian parameters from static and inelastic neutron scattering data, previous approaches have utilized a combination of an autoencoder neural network, used for data compression, and a generative model, used for forward prediction [14, 19–21]. This pipeline has been shown to return excellent results on fully-collected data but has not previously been applied to the setting of on-the-fly parameter extraction.

Prior machine learning efforts in the neutron scattering community have relied on traditional image-based data representations. A promising direction in this field can be capitalized on with the introduction of a new paradigm of data modelling based on neural implicit representations [22, 23]. Such models are often described as coordinate networks as they take a coordinate as input and typically output a single scalar or a small set of scalars. In computational imaging applications, these networks learn mappings from pixel coordinates (i, j) to an RGB value representing the color of that pixel. The coordinate-based representation encodes the image implicitly through a set of trainable weights and can be used to make predictions at sub-pixel scales. These models have been shown to be able to accurately capture high-frequency features in images and scenes and have been particularly successful at tasks such as 3D-shape representation and reconstruction. Furthermore, gradients and higher-order derivatives of the implicit representation can be readily calculated and used for solving inverse-problems [22, 24–26].

In this work, we develop a neural implicit representation for the dynamical structure factor, $S(\mathbf{Q}, \omega)$, as a function of energy transfer ($\hbar\omega$), momentum transfer (\mathbf{Q}), and Hamiltonian parameter coordinates. The dynamical structure factor is a general function measured in many inelastic x-ray and neutron experiments and is related to different correlation functions of the probed order, see Methods for further details. To

demonstrate the versatility of our method, we report the results using a series of calculations based on mean field theory through the linear spin wave theory (LSWT) framework [27]. We simulate LSWT spectra for a spin-1 square-lattice Heisenberg model Hamiltonian over a large phase space of Hamiltonian parameters and use it to train a neural implicit representation. The model is applied to experimental time-of-flight neutron spectroscopy data [28] taken on the quasi-2D Néel antiferromagnet La_2NiO_4 , and leverages a GPU-based optimization procedure to return the Hamiltonian parameters that represent the system under study. In particular, the method does not rely on peak fitting algorithms and performs well under low signal-to-noise ratio scenarios. To gain further insight, we use a Monte-Carlo simulation of the experimental data collection process to demonstrate the potential of our approach for continuous in-situ analysis to provide guidance on when an adequate amount of data has been collected to conclude the experiment. Collectively, these findings pave the way for conducting scattering experiments in a streamlined and efficient manner, and open exciting new avenues to swiftly unravel the parameterization of underlying dynamical models.

RESULTS

Neural Implicit Representation Modelling

Our machine learning framework is based on the concept of implicit neural representations which are machine learning models that can be used to store images (or hypervolumes) via trainable network parameters. Accordingly, we develop a neural implicit representation for the hyper-volume of the dynamical structure factor across different model Hamiltonian parameters. Our Hamiltonian, which corresponds to an extended nearest-neighbor Heisenberg model, is given by Equation 1 [7, 29].

$$\mathcal{H} = J \sum_{\langle i,j \rangle} \hat{\mathbf{S}}_i \cdot \hat{\mathbf{S}}_j + J_p \sum_{\langle i,j' \rangle} \hat{\mathbf{S}}_i \cdot \hat{\mathbf{S}}_{j'}, \quad (1)$$

As depicted in Fig. 1a, J and J_p are the first- and second-nearest-neighbor Heisenberg exchange coupling parameters on a square lattice. Thus, for Q_x and Q_y , a square-lattice notation is utilized with \mathbf{a} and \mathbf{b} corresponding to the vectors connecting the first nearest neighbors or opposing edges of the square, respectively.

The specific implicit neural representation presented in this work is a Sinusoidal Representation Network (SIREN) [22], which is a fully-connected neural network [30] with sinusoidal activation functions that accepts coordinates as input. Our SIREN model is trained to approximate the scalar function $\log(1 + S(\mathbf{Q}, \omega, J, J_p)) \in \mathbb{R}_+^1$ (a real positive number $\{x \in \mathbb{R} | x > 0\}$), which is a logarithmic transformation of the dynamical structure factor evaluated at a specific $\mathbf{Q} \in \mathbb{R}^3$ (3 dimensional, reciprocal lattice vector in reciprocal lattice units (r.l.u.)), $\hbar\omega \in \mathbb{R}^1$ (energy transfer in units of meV) and $J, J_p \in \mathbb{R}^1$ (specific Hamiltonian coupling parameters in units

Figure 1. Overview of machine learning pipeline, model Hamiltonian and reciprocal space paths. **a** Ni_4O_4 square-lattice plaquette in La_2NiO_4 . J (J_p) is the first (second) nearest-neighbor interaction and a and b indicate the square-lattice unit vectors. **b** The Brillouin zone for the spin-1 square-lattice magnetic structure. Selected high-symmetry points are indicated. The two momentum paths are denoted by the purple and orange lines, respectively. **c** Visualization of the SIREN neural network for predicting the scalar dynamical structure factor intensity. All nodes in adjacent layers are connected to each other in a fully-connected architecture. The notation 64×3 and 64×1 , represent three and one neural network layers with 64 neurons and with sinusoidal and linear activation functions, respectively. Neural network bias vectors are omitted for clarity. **d** Visualization of the distribution of training, test, and validation data in J - J_p space. **e** Synthetic $S(\mathbf{Q}, \omega)$ predictions from the SIREN model along the corresponding trajectory shown in **d**. Grid lines correspond to $[0, 50, 100, 150, 200]$ meV and $[\text{P}, \text{M}, \text{X}, \text{P}, \Gamma, \text{X}]$ for the energy and wave vector, respectively.

of meV). We use the logarithm to increase the weighting of weaker features in the data and to prevent ill-conditioned behaviour around zero.

The functional form of the SIREN neural network, denoted as Φ , involves applying a series of matrix multiplications, vector additions and sinusoidal operations to the coordinate vector $[\mathbf{Q}, \omega, J, J_p]^T \in \mathbb{R}^6$ (Equation 2).

$$\begin{aligned}
 h_0 &= \sin(W_0[\mathbf{Q}, \omega, J, J_p]^T + b_0) \\
 h_i &= \sin(W_i h_{i-1} + b_i) \quad \text{with } i \in \{1, 2, 3\} \\
 \Phi &= (W_4 h_3) + b_4
 \end{aligned} \tag{2}$$

Here, $b_0 \in \mathbb{R}^6$, $\{b_1, b_2, b_3\} \in \mathbb{R}^{64}$, $b_4 \in \mathbb{R}^1$, $W_0 \in \mathbb{R}^{64 \times 6}$, $\{W_1, W_2, W_3\} \in \mathbb{R}^{64 \times 64}$ and $W_4 \in \mathbb{R}^{1 \times 64}$ are vectors and matrices, respectively, that are learned during the training process to ensure that $\Phi(\mathbf{Q}, \omega, J, J_p)$ mimics $\log(1 + S(\mathbf{Q}, \omega, J, J_p))$ as closely as possible. Graphically, W_0, W_1, W_2 , and W_3 correspond to the weights between the first four layers of the network which are transformed by applying the sine function in an element-wise manner. W_4 represents the weights for the final layer for which no non-linear function is applied. This specific neural architecture is also illustrated in Fig. 1c.

We note that although our model is written for three-dimensional \mathbf{Q} , the neutron profiles used in the following

sections do not include a Q_z component due to limited sample orientations. The model is trained on 1,200 LSWT simulations of $S(\mathbf{Q}_{\text{list}}, \omega_{\text{list}})$ over a large set of possible J and J_p values and on two paths in reciprocal space (Fig. 1b). \mathbf{Q} -path 1 and 2 are denoted as $\text{P} \rightarrow \text{M} \rightarrow \text{X} \rightarrow \text{P} \rightarrow \Gamma \rightarrow \text{X}$ and $\text{P1} \rightarrow \text{M1} \rightarrow \text{X1} \rightarrow \text{P1} \rightarrow \Gamma1 \rightarrow \text{X1}$ which correspond to $\mathbf{Q}_{\text{path1}} = \{[\frac{3}{4} \frac{1}{4} 0], [\frac{1}{2} \frac{1}{2} 0], [\frac{1}{2} 0 0], [\frac{3}{4} \frac{1}{4} 0], [1 0 0], [\frac{1}{2} 0 0]\}$ and $\mathbf{Q}_{\text{path2}} = [-0.07 \ 0.03 \ 0] + \mathbf{Q}_{\text{path1}}$. Here, $\mathbf{Q}_{\text{list}} \in \mathbb{R}^{N_{\mathbf{Q}}}$ and $\omega_{\text{list}} \in \mathbb{R}^{N_{\omega}}$ is an overloaded notation which refers to a series of $N_{\mathbf{Q}}$ and N_{ω} points in the (\mathbf{Q}, ω) -space, respectively.

Once the differentiable neural implicit model is trained, it is possible to use gradient-based optimization to solve the inverse problem of determining the unknown J and J_p parameters from data. Our objective function for the optimization task measures the Pearson correlation coefficient (r) between $\log(1 + S(\mathbf{Q}, \omega, J, J_p))$ and the machine learning prediction (Equation 3).

$$L = 1 - r(\log(1 + S_{\text{measured}}), \Phi(\mathbf{Q}, \omega, J, J_p)) \quad (3)$$

We use the correlation as the metric because the normalization factors between the experiment and simulation are here unknown. Using the prior is favourable as it enhances the weighting of the coherent excitation at high $\hbar\omega$ and further helps evade contamination due to statistical noise in the elastic and incoherent-inelastic scattering, which arises primarily at low $\hbar\omega$ and that cannot be removed by background subtraction. The latter is important since we are not aiming to fully describe the spectral weights, which would require the exact handling of all individual neutrons in the full three-dimensional \mathbf{Q} -space, instead of the averaged weight in the reduced two-dimensional \mathbf{Q} -space. During optimization, any subset of $(\mathbf{Q}_{\text{list}}, \omega_{\text{list}})$ coordinates can be chosen as long as they fall along either of the paths defined in Fig. 1b. Here, we note that from an inference point of view, any momentum or energy coordinates could be chosen, however our training data only includes two reciprocal-space paths. To determine the Hamiltonian parameters, J and J_p are treated as free parameters in the optimization problem. The objective in Equation 3 is optimized using the Adam optimizer [31], a commonly used gradient-based optimization algorithm that exploits the automatic differentiation capabilities in Tensorflow [32] to calculate $\frac{dL}{dJ}$ and $\frac{dL}{dJ_p}$, see Methods for details.

In our technique, it is not necessary to use all sets of $\mathbf{Q}_{\text{list}}, \omega_{\text{list}}$ along both paths to perform the fitting. Instead, random batches of coordinates $(\mathbf{Q}_{\text{batch}}, \omega_{\text{batch}})$ can be queried at each optimization iteration in order to improve computational efficiency and converge to a better minimum, in a manner similar to the regularization effects of stochastic gradient descent [33]. Pseudo-code for the optimization procedure is provided in Algorithm 1.

Algorithm 1 Differentiable Neural Optimization

```

while N < MaxIter do
   $\mathbf{Q}_{\text{batch}}, \omega_{\text{batch}}, S_{\text{batch}} \sim [\mathbf{Q}_{\text{list}}, \omega_{\text{list}}, S_{\text{list}}]$ 
   $\log(1 + S_{\text{pred}}) = \Phi(\mathbf{Q}_{\text{batch}}, \omega_{\text{batch}}, J, J_p)$ 
   $J, J_p \leftarrow \text{ADAM}(L(S_{\text{batch}}, S_{\text{pred}}))$ 
end while

```

Application to La_2NiO_4

We first characterize the performance of our machine learning framework on simulated SpinW data in order to demonstrate the viability of using a neural implicit representation for the LSWT simulator. Fig. 1e demonstrates the ability of our implicit model to generate new predictions for $S(\mathbf{Q}, \omega)$ under Hamiltonian parameter ranges that lie outside the training data. Fig. 2 provides a comparison between the LSWT and machine learning simulation with specific values of the input parameters ($J = 45.57$ meV and $J_p = 2.45$ meV). In this example, the machine learning framework was fed (J, J_p) directly (instead of obtaining these parameters using gradient descent through the neural representation). The machine learning prediction and the LSWT simulation are seen to be almost indistinguishable. A quantitative analysis of the difference between simulation and prediction is provided in Supplementary Fig. S2.

Figure 2. Comparison between linear spin wave theory simulation and machine learning prediction for a given set of parameter values ($J = 45.57$ meV and $J_p = 2.45$ meV). **a** Example of ground-truth simulated $S(\mathbf{Q}, \omega)$ calculated using the SpinW software program and **b** corresponding machine learning forward model prediction.

Although our model can clearly approximate simulated data well, our main motivation, however, is to provide a tool that can reliably extract the spin Hamiltonian parameters of interest from real, experimental data. For this reason, we applied our method to the measured inelastic neutron scattering data (after an automatic background-subtraction) taken from the quasi-2D Néel antiferromagnet La_2NiO_4 . Experimental data prior to background subtraction are shown in Supplementary Fig. S1. Though a full 3D dataset was collected, we chose two paths in \mathbf{Q} -space to simulate many spectra for a range

Figure 3. Hamiltonian parameter extraction via auto-differentiation of the neural implicit representation. **a** and **b** show experimental data after automated background subtraction. For the experimental data, the color bars reflect $S(\mathbf{Q}, \omega)$ in units of: mbarn sr⁻¹meV⁻¹f.u.⁻¹. **c** and **d** show the corresponding machine learning predictions for the two paths. The predicted profiles are visually seen to be similar the experimental data. Deviations at low $\hbar\omega$ are due to the neglect of anisotropic spin gaps in our model. **e** Visualization of the loss landscape for objective fitting in the Hamiltonian parameter space (J, J_p).

of J and J_p for the model training prior to any inclusion of real data. After the model was trained on the two simulated paths, we applied Algorithm 1 to determine J and J_p from the data. The optimization for both experimental paths was performed jointly, and therefore, the fit parameters are the same for both cases. Our approach was found to yield excellent predictions, both qualitatively and quantitatively, relative to the results of a detailed and expensive analytical fit, as shown in Fig. 3a and b. The analytical parameters in the LSTW limit, adapted from Petsch *et al.* [28], are $J = 29.00(8)$ meV and $J_p = 1.67(5)$ meV. The parameters obtained from our machine learning fitting are $J = 29.68$ meV and $J_p = 1.70$ meV. We also experimented with fitting each path independently and also obtained similar predictions; for path 2, this is especially notable since a significant portion of the experimental data is missing in this case, see Supplementary Fig. S3. Supplementary Fig. S4 provides fitting results from SpinW with algorithmic peak-fitting, which yields similar results for this dataset.

Since our neural implicit model is computationally inexpensive to evaluate, we also constructed a loss landscape of the objective function with respect to J and J_p . The objective function is found to be well-behaved and the gradient descent scheme finds a fit close to the analytical result (Fig. 3e). We emphasize that the only information provided to the algorithm is the knowledge of a region of the $(\mathbf{Q}, \hbar\omega)$ -space on which to carry out an automatic background subtraction prior to fitting the data. Importantly, no peak finding or extraction is needed as the optimization objective uses the intensity of all provided voxels in the $(\mathbf{Q}, \hbar\omega)$ -space or pixels on the 2D intensity map rather than the magnon peak positions $\hbar\omega_{\mathbf{Q}}$.

Real-Time Fitting

In real experimental settings, another critical issue is the ability to make rapid decisions on whether or not sufficient data have been collected at any one time to allow for a good understanding of the physics being explored.

To probe the effectiveness of our framework for real-time fitting during an experiment, and to reduce data collection time, we used the experimental data to generate plausible data for low counting situations. Specifically, we smoothed the experimental data and used it as a probability distribution which is sampled using rejection sampling, see Methods. In a real experiment, a sample is normally measured using a series of different orientations on the spectrometer, often which varying time scales. Here, the rejection sampling simulates the La₂NiO₄ neutron scattering experiment performed in the same sample orientations but with throughout equally shorter data collection times. This exercise gives insight into the viability of our approach for handling low statistics and noisy data. We note that any “detector noise” and scattering from the sample environment is negligible compared to statistical noise in the scattering from the sample. In Fig. 4a, we show the obtained parameters from the machine learning fitting as a function of the number of detected neutrons within the two path regions. Visualizations of path 1 at selected points in time are also shown in Fig. 4b. The machine learning prediction is obtained as the lowest objective value from 10 independent gradient descent optimizations starting from random locations in the Hamiltonian parameter space. Using the median prediction gives very similar results. This test demonstrates that our machine learning model quickly converges to the true solution and is effective under low signal-to-noise conditions.

Figure 4. Real-time Hamiltonian parameter estimation using a differentiable implicit neural representation. **a** Machine learning prediction for J and J_p as a function of the total number of neutrons detected within the two path regions. Square-root scaling is used for the neutron counts due to the Poisson collection statistics. The machine learning prediction is seen to converge much earlier than the count-time recorded in the experiment. **b** Visualization of plausible low-count (without algorithmic background subtraction) data. Total number of neutrons detected with the two path regions are: 16,173, 57,237, and 326,952 (top to bottom). The colorbars show the absolute counts of detected neutrons.

DISCUSSION

In this work, we develop a neural implicit representation customized for inelastic neutron scattering analysis and show that this model can enable precise extraction of Hamiltonian parameters and has the potential to be deployed in real-time settings to minimize required counting time.

We emphasize that our implicit modelling scheme considers data as coordinates $(\mathbf{Q}, \omega, J, J_p)$ which is fundamentally different from the traditional image-based representations. One benefit of this approach is that the model continuously represents energy, momentum, and Hamiltonian parameters, and can therefore be used to make predictions at displaced coordinates $(\mathbf{Q} + \delta\mathbf{Q}, \omega + \delta\omega, J + \delta J, J_p + \delta J_p)$. This enables prediction at finer resolutions of \mathbf{Q} and ω than those recorded on pixelized detectors or at Hamiltonian parameters not present in the training set. Additionally, since the model is a SIREN neural network, it is composed of a series of differentiable operations and is therefore amenable to automatic differentiation techniques. This is highly advantageous and allows the entire analysis pipeline to be compactly expressed by a single model that is end-to-end differentiable relative to the parameters of interest. This approach also allows for an elegant treatment of missing data. Here, missing coordinates can simply be dropped from the parameter estimation step without the need for additional model retraining or data masking.

To validate our approach, we use inelastic scattering data from La_2NiO_4 and find that our method accurately recovers unknown parameters corresponding to the assumed spin-1 Heisenberg Hamiltonian model on a square lattice. The small

overestimation of J arises from several factors. Small differences in the value of J arise from the 3-dimensionality of \mathbf{Q} and the associated variations in the magnetic form factors and polarization factors. Such 3-dimensional information is not included in our analysis since we only consider quantities averaged over $Q_z \in [-10, 10]$ r.l.u. Also, the resolution function and finite lifetime are only approximations here and further, any multi-magnon scattering is not described by LSWT. Finally, we do not include effects of the experimentally observed energy shifts resulting from the spin gaps [28, 34]. These issues could, however, be addressed through more comprehensive simulations within the overall modelling framework presented here.

Another area for improvement concerns the challenging task of background subtraction. For the analysis of La_2NiO_4 , we were able to develop an automatic background subtraction scheme, based on human insight, to successfully suppress non-magnetic contributions which include non-magnetic coherent excitations (phonons here). However, the suppression of other contributions by this method may not always be feasible. In future work – phonon dispersion calculations, nuclear structure factors, and usage of \mathbf{Q} -dependence of spectral weights – could be implemented in our framework to distinguish additional coherent excitations.

Our ability to continuously fit and refine data as it is collected is important for enabling more efficient and informative experimental design. Since neutron scattering measurements typically involve low detector count rates, this is a major factor that will influence the efficiency of measurement time at facilities. Moreover, one would like to minimize the amount

of time needed to complete an experiment without sacrificing data quality. We have shown our model to perform well under low signal conditions and to yield accurate Hamiltonian parameter predictions, thereby providing guidance on when best to conclude data collection. Here, stochastic gradient descent of the neural implicit model is an effective strategy to filter noise and achieve robust optimization. Note that, if other paths in reciprocal space were available, leveraging the information obtained in the additional data would have simply required training with additional simulations, without any necessary changes to the overall machine learning model. This is an important point for real-time applications, as the flexibility of the coordinate-based representation to ingest additional data is a significant advantage over from conventional analysis pipelines, which rely on manually guided peak-fitting algorithms that are not suited to this type of high-dimensional data. We note that the characterization of the framework's effectiveness for real-time fitting only considers the case of shorter counting times across all measured sample orientations, highlighting the framework's capability to handle sparsely distributed detection. Since such measurements usually have to be repeated, this analysis approach could be applied between repetitions to determine whether more data collection is necessary. Furthermore, additional work could involve simulating the training data with respect to sample orientations, which would be preferred when considering experimental guiding for a real, live experiment. In general, we anticipate that our method will be readily compatible with autonomous experimental steering agents by exploiting the model's fast and scalable forward computations which are essential in Bayesian experimental design [35, 36].

Although the present contribution focused on linear spin wave simulations, the approach presented here is not restricted to a particular choice of theoretical scheme. We expect that our framework will be particularly impactful when combined with using expensive and advanced computational methods for simulating strongly correlated systems, such as exact diagonalization (ED) [37], density matrix renormalization group (DMRG) [38, 39], determinant quantum Monte-Carlo (DQMC) [40, 41], and variational Monte Carlo (VMC) [42, 43] simulations.

The methodology presented here breaks the barrier of real-time fitting of inelastic neutron and x-ray scattering data, bypassing the need for complex peak-fitting algorithms or user-intensive post-processing. Our study thus opens new opportunities for significantly improved analysis of excitations in classical and quantum systems.

METHODS

Sample Preparation and Data Collection

In the experiment a 21 g single crystal of the quasi-2D Néel antiferromagnet $\text{La}_2\text{NiO}_{4+\delta}$ ($P4_2/nm$ with $a = b = 5.50 \text{ \AA}$ and $c = 12.55 \text{ \AA}$), grown by the floating-zone technique,

was utilized. The presented time-of-flight neutron spectroscopy data were collected on the SEQUOIA instrument at the Spallation Neutron Source at the Oak Ridge National Laboratory [44] with an incident neutron energy of 190 meV, the high-flux Fermi chopper spun at 300 Hz, and a sample temperature of 6 K. The data is integrated over the out-of-plane momentum $Q_z \in \pm 10 \text{ r.l.u.}$. The lattice can be approximated by $I4/mmm$ with $a = b \approx 3.89 \text{ \AA}$. Q_x and Q_y for $I4/mmm$ are equivalent to Q_x and Q_y in the square-lattice notation. For more details see Ref. [28].

SpinW Simulation and Fitting

In an inelastic scattering experiment the measured quantity is the partial differential cross section which is related to the dynamical structure factor $S(\mathbf{Q}, \omega)$ by $\frac{d^2\sigma}{d\Omega dE_f} = k_f/k_i S(\mathbf{Q}, \omega)$, where k_i and k_f are the incident and final neutron or photon wave vectors. In our simulations, the dynamical structure factor is approximated to $S(\mathbf{Q}, \omega) \propto \sum_{m,n} \int dt e^{-i\mathbf{Q}\cdot(\mathbf{r}_m - \mathbf{r}_n)} e^{-i\omega t} \langle S_m(t) S_n(0) \rangle$, where $\langle S_m(t) S_n(0) \rangle$ represents spin-spin correlations at different atomic sites m, n . The neutron polarization factor as well as the magnetic form factor are neglected here.

The two momentum paths used for $S(\mathbf{Q}, \omega)$ simulation are $\mathbf{Q}_{\text{list1}} = \{ [\frac{3}{4} \frac{1}{4} 0], [\frac{1}{2} \frac{1}{2} 0], [\frac{1}{2} 0 0], [\frac{3}{4} \frac{1}{4} 0], [1 0 0], [\frac{1}{2} 0 0] \}$ and $\mathbf{Q}_{\text{list2}} = [-0.07 \ 0.03 \ 0] + \mathbf{Q}_{\text{list1}}$, respectively in reciprocal lattice units. The SpinW software [12] was used to perform 600 simulations for each path (1200 total) corresponding to randomly sampling J and J_p in ranges of [20, 75] meV and [-30, 10] meV. The lower limit for J and upper limit for J_p are chosen such that the ground state remains the Néel state which is satisfied in LSWT for $J > 2J_p$ and $J > 0$. For each location in \mathbf{Q} , the corresponding energies from 0 - 200 meV were obtained. The quantum fluctuation renormalization factor Z_c is set to 1.09 [28, 45, 46]. After simulation, the data was convoluted with an energy-dependent kernel based on the beamline instrument profile. For this procedure, an in-built tool from SEQUOIA was used to give a polynomial fit for the dependence of the resolution (FWHM) in meV on the energy transfer ($\hbar\omega$) in meV: $\text{FWHM} = 1.4858 \times 10^{-7} (\hbar\omega)^3 + 1.2873 \times 10^{-4} (\hbar\omega)^2 - 0.084492 \hbar\omega + 14.324$ [44]. In addition, the data was broadened with a 1D Gaussian kernel ($\sigma = 5$ pixels) in \mathbf{Q} to correct for the discrete sampling of the simulation and to partially consider the momentum resolution of the instrument.

The SpinW-software-based spin wave spectrum fitting was implemented using its built-in function. The inputs are peak information extracted from experimental spin wave dispersion data. The R value is optimized using a particle swarm algorithm to find the global minimum defined as $R = \sqrt{1/n_E \times \sum_{i,\mathbf{Q}} 1/\sigma_{i,q}^2 (\hbar\omega_{i,\mathbf{Q}}^{\text{sim}} - \hbar\omega_{i,q}^{\text{meas}})^2}$, where (i, q) index the spin wave mode and momentum, respectively. E_{sim} and E_{meas} are the simulated and measured spin wave energies, σ is the standard deviation of the measured spin wave energy determined previously by fitting the inelastic peak and n_E is

the number of energies to fit.

SIREN Model Training

A 5-layer SIREN neural network (Fig. 1c) was trained on 1,000 simulations of $(S(\mathbf{Q}, \omega), J, J_p)$ tuples; 200 simulations were left aside for validation and testing. Here, $\hbar\omega \in [0 - 200]$ meV, $J \in [20 - 75]$ meV and $J_p \in [-30 - 10]$ meV were normalized to 0-1 in order for all the parameters to be on approximately the same scale. The model was trained to predict $\log(1 + S(\mathbf{Q}, \omega, J, J_p))$ by optimizing the mean-squared-error objective L between the prediction and the label with respect to the network parameters. During training, the following hyperparameters and settings were used: Adaptive Moment Estimation (ADAM) algorithm for optimization ($\beta_1 = 0.9$, $\beta_2 = 0.999$) [31], batch size = 2,048, learning rate = 0.001. Here, β_1 and β_2 influence the degree to which past gradients affect the current step. The batch size is a parameter which controls the number of images used to compute the mean-squared-error objective and the learning rate controls the gradient descent step size. The learning rate was exponentially decayed by a factor of $\exp(-0.1)$ for every epoch (full pass through the entire dataset) after the first ten epochs. We used NVIDIA A100 GPU hardware with the Keras API [47] and the model was trained for 50 epochs.

Machine Learning Parameter Extraction

Prior to differentiable optimization, the experimental data were automatically background subtracted using the following procedure. First, a region of $(\mathbf{Q}_{\text{list}}, \omega_{\text{list}})$ space was chosen for each slice (160-170 pixel location in the \mathbf{Q} -axis) and averaged across \mathbf{Q}_{list} to yield a one-dimensional energy profiles. This procedure was chosen based on prior assumptions on the isotropic nature of the scattering and the Néel ground state. Next, the one-dimensional energy profiles were fit using a Savitzky-Golay filter (window size = 51, polynomial order = 3) and used for background subtraction.

The unknown J and J_p parameters were recovered from data using gradient-based optimization of the neural network implicit representation. For the experimental data presented in this work, the metric $(1 - r)$ between the measured and simulated $(1 + S(\mathbf{Q}, \omega, J, J_p))$ was used as the objective function L introduced in Equation 3; here, r refers to the Pearson correlation coefficient. No normalization was performed for scaling the simulation data relative to the experimental data.

The objective L was optimized using the ADAM algorithm with respect to J and J_p and \mathbf{Q}_{list} and ω_{list} were randomly sampled from the list of paths containing the experimental data. Here, a batch size of 4,096 was used for the $(\mathbf{Q}_{\text{list}}, \omega_{\text{list}})$ sampling, with 2,000 Adam optimization steps and a learning rate of 0.005. Here, the batch size refers to the number of pixels in the experimental image that are randomly selected in each step of the optimization procedure.

Low count data generation and fitting

High-count data for each slice (without background subtraction) were smoothed using a 3x3 Gaussian convolutional kernel. The resultant images were each normalized to (0, 1) using the total intensity. Each slice was treated as a probability distribution which was sampled using Monte-Carlo rejection sampling. This process was used to create a series of datasets with neutron counts in the range $(1 \times 10^4 - 9 \times 10^6)$. Each dataset was individually and automatically background subtracted by the previously described method and fit ten times from random starting locations in (J, J_p) using the machine learning optimization procedure. Note, the corresponding low-count data was used in order to perform the automated background subtraction.

DATA AVAILABILITY

All data generated in this study as well as a minimal dataset have been deposited in the Zenodo database under accession code 10.5281/zenodo.8267499 [49].

CODE AVAILABILITY

The code developed in this study have been deposited in the Zenodo database under accession code 10.5281/zenodo.8267474 [48] and is also available at <https://github.com/slaclab/neural-representation-sqw.git>.

AUTHOR CONTRIBUTIONS

S. R. C, Z. J. and A. N. P. contributed equally to this work and focused on the machine learning, simulation and experimental portions respectively. A. I. K. and S. M. H. assisted with data collection. C. P., Z. C., R. P., H. C., S. M., M. D., S. C., A. B., A. F., D. P. and D. R. assisted with data analysis and manuscript writing. C. J., Y. N. and J. T. supervised the work.

COMPETING INTERESTS

The authors declare no competing interests.

ACKNOWLEDGEMENTS

This work is supported by the U.S. Department of Energy, Office of Science, Basic Energy Sciences under Award No. DE-SC0022216, as well as under Contract DE-AC02-76SF00515, both for the Materials Sciences and Engineering Division, as well as for the Linac Coherent Light Source

(LCLS), part of the Scientific User Facilities Division. A portion of this research used resources at the Spallation Neutron Source, a DOE Office of Science User Facility operated by the Oak Ridge National Laboratory. J. J. Turner acknowledges support from the U.S. DOE, Office of Science, Basic Energy Sciences through the Early Career Research Program. Z. Ji is supported by the Stanford Science fellowship, and the Urbenek-Chodorow postdoctoral fellowship awards. A.N. Petsch and S.M. Hayden acknowledge funding and support from the Engineering and Physical Sciences Research Council (EPSRC) under Grant Nos. EP/L015544/1 and EP/R011141/1.

* chitturi@stanford.edu

† zhurun@stanford.edu

‡ apetsch@stanford.edu

§ joshuat@slac.stanford.edu

- [1] Chumak, A. V., Vasyuchka, V. I., Serga, A. A. & Hillebrands, B. Magnon spintronics. *Nature Physics* **11**, 453–461 (2015).
- [2] Neusser, S. & Grundler, D. Magnonics: Spin Waves on the Nanoscale. *Advanced Materials* **21**, 2927–2932 (2009).
- [3] Gutfleisch, O. *et al.* Magnetic materials and devices for the 21st century: stronger, lighter, and more energy efficient. *Advanced Materials* **23**, 821–842 (2011).
- [4] Rossat-Mignod, J. *et al.* Neutron scattering study of the $\text{YBa}_2\text{Cu}_3\text{O}_{6+x}$ system. *Physica C: Superconductivity* **185**, 86–92 (1991).
- [5] Chatterji, T. *Neutron Scattering from Magnetic Materials* (Elsevier, 2005).
- [6] Braden, M. *et al.* Inelastic neutron scattering study of magnetic excitations in Sr_2RuO_4 . *Physical Review B* **66**, 064522 (2002).
- [7] Coldea, R. *et al.* Spin Waves and Electronic Interactions in La_2CuO_4 . *Physical Review Letters* **86**, 5377–5380 (2001).
- [8] Weinfurter, K., Mattingly, J., Brubaker, E. & Steele, J. Model-based design evaluation of a compact, high-efficiency neutron scatter camera. *Nuclear Instruments and Methods in Physics Research Section A: Accelerators, Spectrometers, Detectors and Associated Equipment* **883**, 115–135 (2018).
- [9] Peterson, P. F., Olds, D., Savici, A. T. & Zhou, W. Advances in utilizing event based data structures for neutron scattering experiments. *Review of Scientific Instruments* **89**, 093001 (2018).
- [10] Chen, Z. *et al.* Machine learning on neutron and x-ray scattering and spectroscopies. *Chemical Physics Reviews* **2**, 031301 (2021).
- [11] Li, Z., Kermodé, J. R. & De Vita, A. Molecular Dynamics with On-the-Fly Machine Learning of Quantum-Mechanical Forces. *Physical Review Letters* **114**, 096405 (2015).
- [12] Toth, S. & Lake, B. Linear spin wave theory for single-Q incommensurate magnetic structures. *Journal of Physics: Condensed Matter* **27**, 166002 (2015).
- [13] Ewings, R. *et al.* HORACE: Software for the analysis of data from single crystal spectroscopy experiments at time-of-flight neutron instruments. *Nuclear Instruments and Methods in Physics Research Section A: Accelerators, Spectrometers, Detectors and Associated Equipment* **834**, 132–142 (2016).
- [14] Doucet, M. *et al.* Machine learning for neutron scattering at ORNL. *Machine Learning: Science and Technology* **2**, 023001 (2020).
- [15] Butler, K. T., Le, M. D., Thiyaalingam, J. & Perring, T. G. Interpretable, calibrated neural networks for analysis and understanding of inelastic neutron scattering data. *Journal of Physics: Condensed Matter* **33**, 194006 (2021).
- [16] Anker, A. S., Butler, K. T., Le, M. D., Perring, T. G. & Thiyaalingam, J. Using generative adversarial networks to match experimental and simulated inelastic neutron scattering data. *Digital Discovery* (2023).
- [17] Wang, H. *et al.* Rapid Identification of X-ray Diffraction Patterns Based on Very Limited Data by Interpretable Convolutional Neural Networks. *Journal of Chemical Information and Modeling* **60**, 2004–2011 (2020).
- [18] Chitturi, S. R. *et al.* Automated prediction of lattice parameters from X-ray powder diffraction patterns. *Journal of Applied Crystallography* **54**, 1799–1810 (2021).
- [19] Samarakoon, A. M. *et al.* Machine-learning-assisted insight into spin ice $\text{Dy}_2\text{Ti}_2\text{O}_7$. *Nature Communications* **11**, 1–9 (2020).
- [20] Samarakoon, A., Tennant, D. A., Ye, F., Zhang, Q. & Grigera, S. A. Integration of machine learning with neutron scattering for the Hamiltonian tuning of spin ice under pressure. *Communications Materials* **3**, 1–11 (2022).
- [21] Samarakoon, A. M. *et al.* Extraction of interaction parameters for $\alpha\text{-RuCl}_3$ from neutron data using machine learning. *Physical Research* **4**, L022061 (2022).
- [22] Sitzmann, V., Martel, J., Bergman, A., Lindell, D. & Wetzstein, G. Implicit Neural Representations with Periodic Activation Functions. *Advances in Neural Information Processing Systems* **33**, 7462–7473 (2020).
- [23] Xie, Y. *et al.* Neural Fields in Visual Computing and Beyond. *Computer Graphics Forum* **41**, 641–676 (2022).
- [24] Cheong, S. *et al.* Novel light field imaging device with enhanced light collection for cold atom clouds. *Journal of Instrumentation* **17**, P08021 (2022).
- [25] Vlašić, T., Nguyen, H. & Dokmanić, I. Implicit Neural Representation for Mesh-Free Inverse Obstacle Scattering. *arXiv preprint* (2022). arXiv:2206.02027.
- [26] Levy, A., Wetzstein, G., Martel, J. N., Poitevin, F. & Zhong, E. Amortized Inference for Heterogeneous Reconstruction in Cryo-EM. *Advances in Neural Information Processing Systems* **35**, 13038–13049 (2022).
- [27] Kubo, R. The Spin-Wave Theory of Antiferromagnetics. *Physical Review* **87**, 568 (1952).
- [28] Petsch, A. N. *et al.* High-energy spin waves in the spin-1 square-lattice antiferromagnet La_2NiO_4 . *arXiv preprint* (2023). arXiv:2304.02546.
- [29] Marshall, W. & Lovesey, S. W. *Theory of Thermal Neutron Scattering* (Oxford University Press, 1971).
- [30] Hastie, T., Tibshirani, R., Friedman, J. H. & Friedman, J. H. *The Elements of Statistical Learning: Data Mining, Inference, and Prediction*, vol. 2 (Springer, 2009).
- [31] Kingma, D. P. & Ba, J. Adam: A Method for Stochastic Optimization. *arXiv preprint* (2014). arXiv:1412.6980.
- [32] Abadi, M. *et al.* Tensorflow: Large-Scale Machine Learning on Heterogeneous Distributed Systems. *arXiv preprint* (2016). arXiv:1603.04467.
- [33] Bottou, L. Stochastic Gradient Descent Tricks. *Neural Networks: Tricks of the Trade: Second Edition* 421–436 (2012).
- [34] Nakajima, K., Yamada, K., Hosoya, S., Omata, T. & Endoh, Y. Spin-Wave Excitations in Two Dimensional Antiferromagnet of Stoichiometric La_2NiO_4 . *Journal of the Physical Society of Japan* **62**, 4438–4448 (1993).
- [35] Granade, C. E., Ferrie, C., Wiebe, N. & Cory, D. G. Robust online Hamiltonian learning. *New Journal of Physics* **14**, 103013

- (2012).
- [36] McMichael, R. D. & Blakley, S. M. Simplified Algorithms for Adaptive Experiment Design in Parameter Estimation. *Physical Review Applied* **18**, 054001 (2022).
 - [37] Dagotto, E. Correlated electrons in high-temperature superconductors. *Reviews of Modern Physics* **66**, 763 (1994).
 - [38] White, S. R. Density matrix formulation for quantum renormalization groups. *Physical Review Letters* **69**, 2863–2866 (1992).
 - [39] White, S. R. Density-matrix algorithms for quantum renormalization groups. *Physical Review B* **48**, 10345 (1993).
 - [40] Blankenbecler, R., Scalapino, D. J. & Sugar, R. L. Monte Carlo calculations of coupled boson-fermion systems. I. *Physical Review D* **24**, 2278–2286 (1981).
 - [41] White, S. R. *et al.* Numerical study of the two-dimensional Hubbard model. *Phys. Rev. B* **40**, 506–516 (1989).
 - [42] Ferrari, F., Parola, A., Sorella, S. & Becca, F. Dynamical structure factor of the $J_1 - J_2$ Heisenberg model in one dimension: The variational Monte Carlo approach. *Physical Review B* **97**, 235103 (2018).
 - [43] Hendry, D., Chen, H., Weinberg, P. & Feiguin, A. E. Chebyshev expansion of spectral functions using restricted Boltzmann machines. *Physical Review B* **104**, 205130 (2021).
 - [44] Granroth, G. E. *et al.* SEQUOIA: A Newly Operating Chopper Spectrometer at the SNS. *Journal of Physics: Conference Series* **251**, 012058 (2010).
 - [45] Igarashi, J.-i. $1/S$ expansion for thermodynamic quantities in a two-dimensional Heisenberg antiferromagnet at zero temperature. *Physical Review B* **46**, 10763 (1992).
 - [46] Singh, R. R. Thermodynamic parameters of the $T=0$, spin-1/2 square-lattice Heisenberg antiferromagnet. *Physical Review B* **39**, 9760 (1989).
 - [47] Chollet, F. *et al.* Keras. <https://keras.io> (2015).
 - [48] Chitturi, S. R. *et al.* Code accompanying: Capturing dynamical correlations using implicit neural representations. Zenodo. 10.5281/zenodo.8267474. [neural-representations-sqw. https://doi.org/10.5281/zenodo.8267474](https://doi.org/10.5281/zenodo.8267474)
 - [49] Chitturi, S. R. *et al.* Dataset: Capturing dynamical correlations using implicit neural representations. Zenodo. 10.5281/zenodo.7804447. <https://doi.org/10.5281/zenodo.8267499>

Supplementary Information for Capturing dynamical correlations using implicit neural representations

Figure S1. Inelastic neutron scattering dataset without automated background subtraction. Visualization of **a** path 1 and **b** path 2 for inelastic neutron scattering data without automated background subtraction (Fig. S1). Note, path 2 contains a portion of missing data.

Figure S2. Visualization of the numerical subtraction between LSWT simulation and machine learning forward model prediction. (a) Difference plot with minimum and maximum colorbar limits set at the maximal positive and negative deviation and corresponding to the calculation in Fig. 2. Note, there appears to be some alternating patterns in the difference profile. This is likely due to the discrete choice of momenta simulated with SpinW which are smoothed out by the continuous machine learning prediction. **In regions outside of the dispersion curve, the difference is 0 since the model is able to confidently predict that there is no signal.**

Figure S3. Machine learning forward model accurately predicts scattering profile for a missing region (red rectangle) of path 2 data. For the displayed path there is no data available in the utilized neutron dataset for the missing region of \mathbf{Q} -space. However, data is available for $(Q_x, Q_y) \mapsto (Q_y, Q_x)$, so for the 2D momenta of the missing path region rotated by 45° . As this compound is assumed to be tetragonal magnetically ordered $S(Q_x, Q_y, \omega) = S(Q_y, Q_x, \omega)$ and thus, the missing region in \mathbf{Q} can be substituted by the equivalent data with Q_x and Q_y exchanged. The data along the full path with the missing region substituted is depicted in comparison with the result predicted by our forward model. **a** Experimental data with missing region filled in by exchanging Q_x and Q_y . **b** machine learning prediction using only experimental data from path 2 with missing region. Evidently, the machine learning prediction closely models the true experimental data. Note, in this case, the prediction for path 2 only uses the data from path 2. No information from path 1 is utilized in the fitting.

Figure S4. SpinW fitting results for the (J, J_p) parameters when both path 1 and path 2 are used for the fitting. SpinW optimization yielded Hamiltonian parameters $(J, J_p) = (29.15, 1.55)$ meV. Corresponding forward predictions for **a** path 1 and **b** path 2. Relative to the machine learning method, SpinW fitting does not utilize all the available pixel information and instead requires additional peak finding and peak fitting steps.